# *NXPE1* alters the sialoglycome by acetylating sialic acids in the human colon

Bum Seok Lee [1], Ashley Cook[1,2], Surojit Sur [1,3], Laura Dobbyn[3], Maria Popoli[1], Sana Khalili [4], Shibin Zhou [1,2,3], Chetan Bettegowda [1,5], Nickolas Papadopoulos [1,2,3], Kathy Gabrielson[6], Phillip Buckhaults [4], Bert Vogelstein [1,2,3,7,8], Kenneth W. Kinzler [1,2,3] ✉ & Nicolas Wyhs [1,3] ✉

Mild periodic acid Schiff staining (mPAS) of human colonic tissue has been used to answer a variety of fundamental questions in germline and somatic genetics. mPAS stains sialic acids except when these glycans are modified by O-acetylation, but a full accounting of the genes contributing to sialoglycan acetylation is incomplete. Using haplotypes derived from whole genome sequencing, we identify a region on chromosome 11 that is associated with inherited differences in mPAS staining. Of the genes in this region, only haplotypes containing *NXPE1* correlate perfectly with mPAS staining in the original cohort used for whole genome sequencing, as well as in a validation cohort. Transcriptomic analysis indicates that linked haplotypes are associated with altered expression of *NXPE1* suggesting a possible genetic mechanism. Genetic manipulation of a common single nucleotide polymorphism observed in the haplotype region and located in *NXPE1*'s promoter alters expression and causes changes to modified sialic acid levels supporting this mechanism. Finally, high-performance liquid chromatography (HPLC) confirms that enzymatically active NXPE1 is capable of transferring an acetyl group from acetyl coenzyme A to sialic acid in vitro. These findings suggest that *NXPE1* is the long-sought gene responsible for differences in colon mPAS staining and may be the prototype of a new family of sialic acid O-acetylation-modifying genes.

Since 1946, pathologists have used periodic acid Schiff staining (PAS) to look for mucins—proteins that are heavily glycosylated and are a large constituent of mucus—in various human tissues[1]. Major constituents of the terminal end of glycosylated proteins, as well as glycosylated lipids, are sialic acids which, in turn, are post-translationally modified and together constitute the sialoglycome[2–4]. Such modifications are important as they can abrogate or encourage binding to other cellular constituents depending on the context of the interaction in

question[3]. This impacts a number of cellular processes, including host-pathogen interactions and cell differentiation during development[5]. A variant of PAS staining, called mild periodic acid Schiff staining (mPAS), can distinguish between O-acetylated (mPAS negative) and non-O-acetylated (mPAS positive) sialic acids in tissues, including the colon[5–15]. Previous studies have demonstrated that mPAS staining patterns follow Hardy-Weinberg patterns in normal human colon tissue. Moreover, the number of sporadically positive crypts in otherwise

[1]Ludwig Center for Cancer Genetics and Therapeutics, Johns Hopkins University School of Medicine, Baltimore, MD, USA. [2]Cellular and Molecular Medicine Graduate Program, Johns Hopkins University School of Medicine, Baltimore, MD, USA. [3]Department of Oncology, Johns Hopkins Medical Institutions, Baltimore, MD, USA. [4]College of Pharmacy, University of South Carolina, Columbia, SC, USA. [5]Department of Neurosurgery, Johns Hopkins University School of Medicine, Baltimore, MD, USA. [6]Department of Molecular and Comparative Pathobiology, Johns Hopkins University School of Medicine, Baltimore, MD, USA. [7]Howard Hughes Medical Institute, Johns Hopkins University School of Medicine, Baltimore, MD, USA. [8]Sol Goldman Pancreatic Cancer Research Center, Johns Hopkins University School of Medicine, Baltimore, MD, USA. ✉e-mail: kinzlke@jhmi.edu; nwyhs1@jhmi.edu

negatively staining individuals increases with age, location within the colon, a history of radiation therapy, and chronic inflammatory states[11,16–20]. These somatically acquired changes in staining have been used to shed light on a number of fundamental questions, including the nature of stem cells and the rate of somatic mutation in normal colonic epithelium[21–25]. Uniform positive mPAS staining varies with racial background, with individuals of Asian descent staining positively more frequently than those of European or African descent[18]. Inbred murine models do not demonstrate variable staining, and are uniformly mPAS positive[26,27].

These observations support the idea that an unknown autosomal dominant human gene is responsible for the differential O-acetylation that underlies mPAS staining[17,18]. To date, the only known human sialic acid O-acetyltransferase (SOAT) is encoded by the gene *CASD1*, but its correlation with human colon mPAS staining has not been investigated[28,29]. Given that acetylation modification occurs at many sites on sialic acid in different tissues, several other genes that control O-acetylation likely remain to be discovered[3,30,31].

Here, we describe studies suggesting that one such gene is neurexophilin and PC-esterase domain family member 1 (*NXPE1*). Using a combination of genome-wide association studies and biochemistry, we establish that a haplotype on chromosome 11 is responsible for the mPAS staining phenotype in colorectal tissue. Using genetic manipulation, we confirm that *NXPE1* is the gene causing this phenotype and that recombinant protein is able to acetylate sialic acids in vitro.

## Results

### Whole genome sequencing identifies a haplotype on chromosome 11 associated with mPAS staining

Whole genome sequencing was performed on DNA from the colonic tissue of 21 normal individuals: 8 were negative by mPAS staining, 8 were positive, and another 5 were negative with at least one positive crypt among thousands of non-staining crypts. Such patients were presumed to be of heterozygous genotype for the unknown gene of interest (Fig. 1A). The purpose of sequencing was to identify single-nucleotide polymorphisms (SNPs) or small insertions or deletions (indels) that correlated with mPAS staining (Supplementary Data 1). mPAS staining is primarily localized to goblet cells within normal colorectal crypts (Fig. 1A). An initial genome-wide review identified 202 SNPs with two or fewer genotype/phenotype mismatches in the 21 patient samples examined, with chromosome 11 harboring 90 such SNPs, far more than any other chromosome (Fig. 1B and Supplementary Fig. 1, Supplementary Data 2). A more stringent analysis revealed that all 17 SNPs that showed perfect concordance with the mPAS phenotype were on chromosome 11 and enriched in a small region on chromosome 11q23.2 (chr11:114,298,921 to chr11:114,446,104) (orange bars in Fig. 1B, and Supplementary Fig. 1). An expanded illustration of this 173 kb region is shown in Fig. 1C and Supplementary Fig. 2, and shows a strong linkage disequilibrium ($r^2$ values > 0.8) around the promoter region of the gene *NXPE1*.

### Targeted sequencing implicates *NXPE1* as the gene most tightly associated with colorectal sialic acid mPAS staining status

Four protein coding genes reside in the region described above: *NXPE1*, neurexophilin and PC-esterase domain family member 4 (*NXPE4*), RNA binding motif protein 7 (*RBM7*) and RNA exonuclease 2 (*REXO2*). A review of the 17 perfectly correlated SNPs showed none were in coding regions, four were upstream of either *REXO2* or *NXPE1*, and the others were in intergenic or intronic regions, or downstream, of one of the four genes (Supplementary Fig. 2). To expand on these results, we obtained 91 new samples of normal colonic tissue and stained them to assess mPAS status. Targeted sequencing of the four SNPs upstream of *REXO2* and *NXPE1* showed that only one of them, rs661946, maintained a perfect match (91/91) between the observed phenotypic staining by mPAS and the genotype (Fig. 1D and

Supplementary Data 3). This SNP was only 6 bp upstream of the transcriptional start site of *NXPE1*, presumably part of its promoter, and exists as part of or immediately adjacent to at least three transcription factor binding motifs (ETS-2, SOX1 and GR-α)[32–34]. The observed frequencies for this biallelic SNP (C and T at 53% and 47%, respectively) are consistent with Hardy-Weinberg equilibrium (Haldane's Exact Test). They are also consistent with the published minor allele frequency (MAF) of mPAS positive staining for individuals of East Asian descent, the location from which most of the 91 samples originated[18,35]. The only other SNP significantly associated with mPAS staining was rs561722 (*NXPE2P1*), but it still contained 3/91 mismatches (Supplementary Data 3).

### Structural similarities of *NXPE1* to genes known to O-acetylate sialic acid

*NXPE1* and *NXPE4* have been reported as containing secondary structural similarities to known sialic acid O-acetyltransferases from other species[36]. *NXPE1* contains two motifs, Gly-Asp-Ser (GDS) and Asp-X-X-His (DXXH), with identical amino acid sequence to the catalytic site of the only known human SOAT, the protein CASD1 (expect value = 0.004, 23.9% identity)(Fig. 2A, B). Like CASD1, NXPE1 protein lacks the canonical glycine and asparagine in the GDSL/SGNH fold present in SOATs from other species such as rhamnogalacturonan acetylesterase (Q00017) and photobacterium sp. J15 (AKQ62669.1), making it part of the GDSL/SGNH-like acyl-esterase family[29,36]. That said, the catalytic sites of NXPE1 do contain similarities to the viral homologs of SOAT proteins found in influenza C/JHG/66 virus, Isavirus salaris, bovine torovirus strain Breda 2, and human coronavirus OC43 (Fig. 2A, B)[29,37,38]. These sequence similarities are also conserved among other members of the *neurexophilin/NXPE* family including *NXPE4*, though they display a different, His-Pro-Pro, sequence for their histidine containing motif (Fig. 2B)[36,39]. In silico 3D structure prediction of NXPE1 colocates Asp526 and His529 adjacent to the Ser355, creating a hydrogen bond between the histidine and serine residues similar to the catalytically active serine (Ser94) in CASD1 (Fig. 2C, D, and Supplementary Fig. 3)[40].

### NXPE1 protein levels correlate with sialic acid mPAS status and to the staining of sialic acid binding lectins

We developed immunohistochemical (IHC) assays for NXPE1, NXPE4, and CASD1 (the only known human SOAT) and evaluated their expression in colonic tissue. All three proteins showed a membrane granular immunostaining pattern limited to colon epithelium. This is consistent with RNA and protein expression data noted in publicly available databases and in the limited number of publications that discuss NXPE1, which describe its importance in gastrointestinal health[41–43]. To facilitate understanding of the NXPE1 staining patterns, Supplementary Data 4 provides the results expected from the major hypotheses evaluated, both for proteins and the other assays described later in this study. Tissues with a homozygous T variant of the rs661946 SNP displayed no NXPE1 protein (by IHC) and robust mPAS staining, while heterozygous samples showed NXPE1 immunolabeling but no mPAS staining (Fig. 3A and Supplementary Fig. 4, Supplementary Data 4). Most impressively, some samples with no mPAS staining in the vast majority of crypts contained a crypt or a discrete segment of a crypt that stained robustly, presumably representing focal loss of sialic acid modification. Patient tissue samples displaying this staining pattern always had a heterozygous (C/T) genotype of the rs661946 SNP, and the rare mPAS staining crypts (less than 1 out of a 1000 crypts) always displayed total loss of NXPE1 protein (Figs. 3A, B and Supplementary Fig. 4, Supplementary Data 4). In contrast, NXPE4 and CASD1 staining was independent of mPAS staining (Fig. 3B).

We next sought independent evidence that altered mPAS staining was due to modifications of sialic acid. It is established that the sialic acid-binding immunoglobulin-like lectin (*SIGLEC*) family binds to

sialylated glycoproteins[44–47]. Moreover, O-acetylation modifications are known to modulate sialic acid-SIGLEC interactions for some members of the SIGLEC family[48,49]. Of twenty SIGLEC/sialic acid binding proteins tested, sialic acid-binding Ig-like lectin 15 (SIGLEC-15) consistently stained colon tissues in the same way as mPAS, suggesting a previously unknown role of O-acetylation modifications for SIGLEC-15 binding (Figs. 3C, D, Supplementary Data 4). Importantly, SIGLEC-15 staining recapitulated the rare sporadic clusters of NXPE1 negative and

mPAS positive crypts in samples heterozygous for the rs661946 SNP (Figs. 3C, D). SIGLEC-15 is a lectin known to bind Ser/Thr N-acetylgalactosamine (O-GalNAc)-bound sialic acids, and also binds with lower affinity to N-acetylglucosamine (GlcNAc)-bound sialic acids[50,51]. Functionally, SIGLEC-15 has been described as a regulator of osteoclast differentiation as well as a potential immune checkpoint for macrophages[52,53]. SIGLEC-7 also showed some specificity for mPAS positive colon tissue, but the signal was not as consistent or uniform as

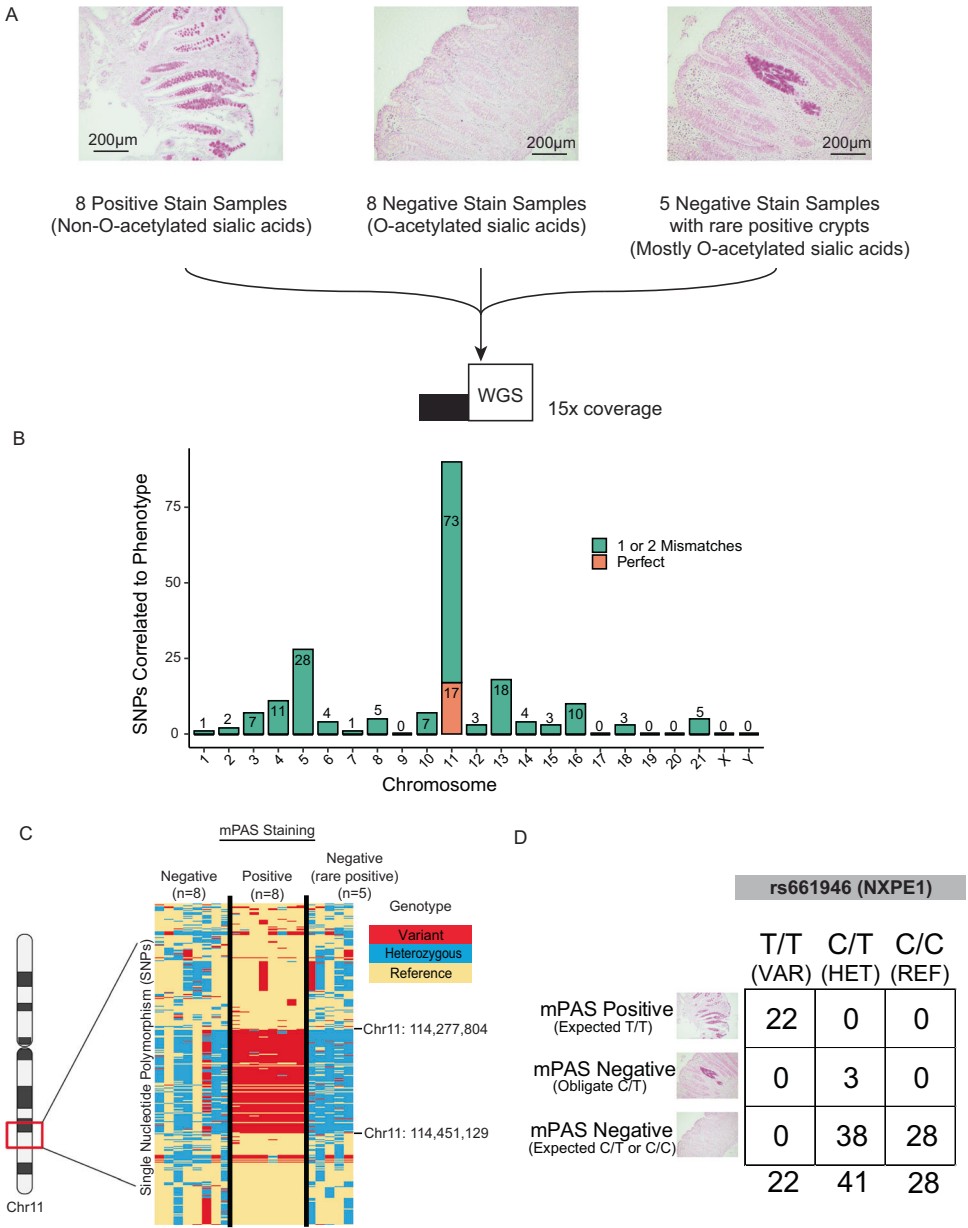

**Fig. 1 | WGS identifies haplotype on chromosome 11 associated with colorectal sialic acid acetylation status based on mPAS staining. A** Schematic showing samples and NGS strategy employed to identify regions associated with colon mPAS staining. Negative staining samples with a rare positive crypt were presumed to be of heterozygous genotype, where a spontaneous loss of heterozygosity in a stem cell caused positive staining of a crypt (right). Images shown are 10x. **B** Bar plot showing the number of SNPs correlated with the mPAS phenotype by chromosome as evaluated by WGS (green shading). Only SNPs on chromosome 11 contained perfect matches between genotype and phenotype in all samples (orange shading). **C** Region on chromosome 11q23.2 identified by whole genome sequencing as highly associated with mPAS staining. Samples that largely stain

negative but contain rare positive crypts are suspected of having a heterozygous genotype and are listed separately (blue shading). Homozygous reference, heterozygous and homozygous alternate genotypes are noted by yellow, blue and red shading respectively. SNPs are shown on the y-axis based on their location on chromosome 11. Formal linkage disequilibrium analysis of this region is shown in figure S2. **D** Independent validation of WGS results. 3 × 3 table showing genotype for SNP rs661946 (located in the promoter of NXPE1) and mPAS staining phenotype on a set of 91 normal colon tissue samples. Allele frequencies are consistent with Hardy-Weinberg equilibrium (Haldane Exact = 0.45). A two-sided Fisher's Exact Test for the genotype-phenotype relationship yields a $p < 2.2e{-}16$.

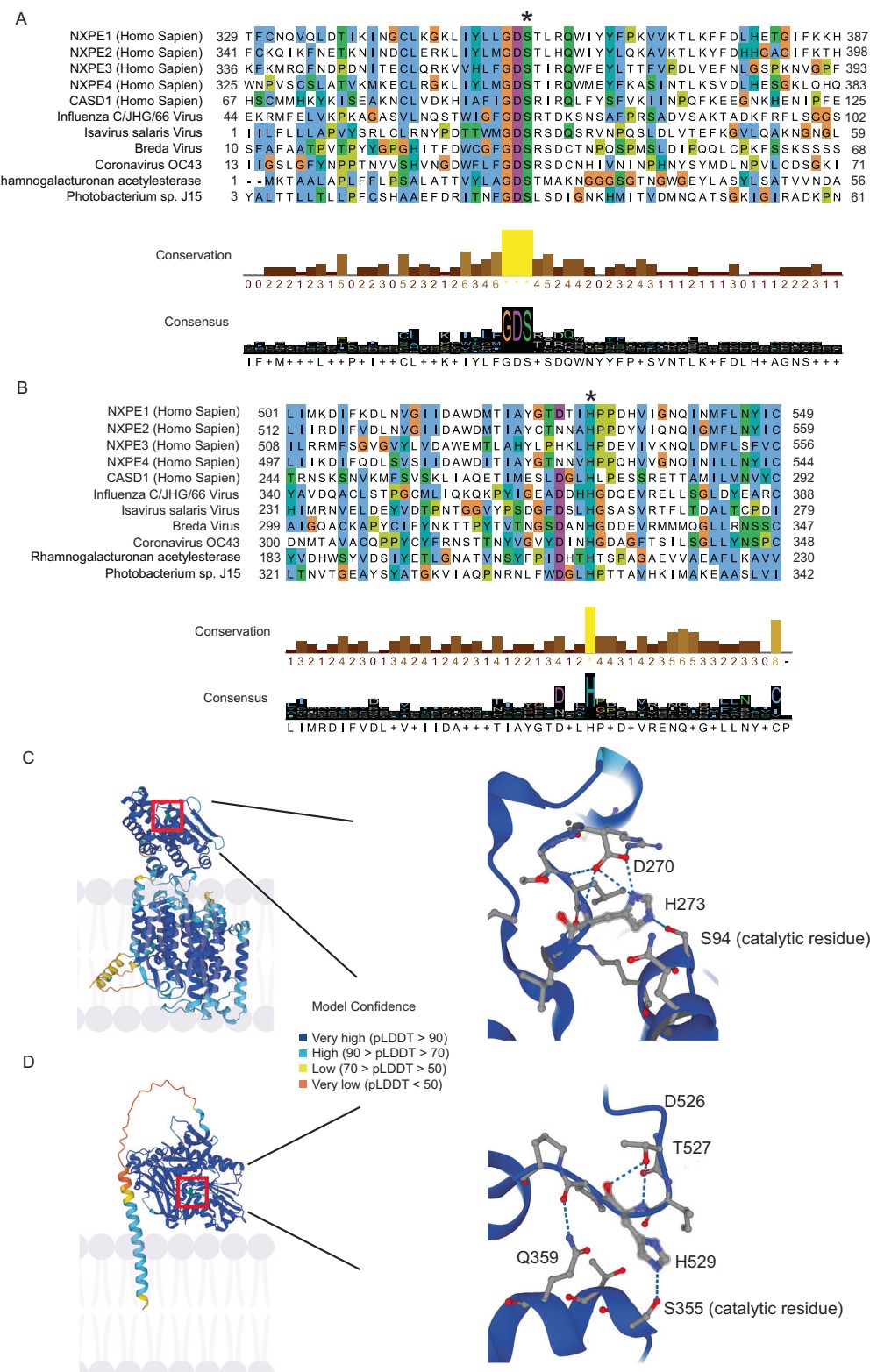

**Nature Communications** | (2025)16:4912

that of SIGLEC-15 (Supplementary Fig. 5). This observation is supported by previous evidence showing that SIGLEC-7 and SIGLEC-15 bind to similar clusters of O-glycan bound sialic acids[50].

## Manipulation of NXPE1 protein expression alters sialic acid modification

Using a lentivirus, we introduced a *NXPE1* expression cassette under the control of a Cytomegalovirus (CMV) promoter into Jurkat cells derived from a human T-cell leukemia patient. Jurkat was chosen as it expressed sialyl-Tn, a truncated sialic acid-containing O-glycan known to bind to SIGLEC-15 and commonly found on cancer cells, and has very low expression of *NXPE1* (Supplementary Fig. 6)[54]. In a pool of cells infected with a lentivirus containing a *NXPE1* open reading frame, increased amounts of NXPE1 protein led to decreased SIGLEC-15 binding by flow cytometry, consistent with the hypothesis that increased *NXPE1*

**Fig. 2 | NXPE1 protein sequence and 3D structure have close homology to known sialic acid-O-acetyltransferases.** *NXPE1* and *NXPE4* are part of a small family of proteins known as neurexophilin/NXPE, with 4 known members. The *NXPE1* gene codes for a 547 amino acid protein predicted to contain a N-terminal transmembrane domain, with the majority of the protein residing on the extracellular portion of the cell. **A** Multiple sequence alignment of amino acid sequences near the serine and (**B**) aspartate-histidine catalytic sites of NXPE family members with known sialic acid-O-acetyltransferases. NXPE1 (Q8N323.2) was compared to human proteins neurexophilin and PC-esterase domain family member 2 (NXPE2) (Q96DL1.2), neurexophilin and PC-esterase domain family member 3 (NXPE3) (Q969Y0.1), NXPE4 (Q6UWF7.1) and CASD1 (Q96PB1.1), as well as viral and bacterial

SOAT proteins from influenza C virus (C/Johannesburg/1/66) (P07975.1), Isavirus salaris (AAL34465.1), Breda virus (CAA71819.1), Human coronavirus OC43 (P30215.1), and known SGNH/GDSL hydrolase family members rhamnogalacturonan acetylesterase (Q00017), photobacterium sp. J15 (AKQ62669.1). The active site residues are indicated with an *. Conservation shown above is a quantification of the alignment of residues as determined by pre-defined physico-chemical properties. Consensus refers to the percentage of which the most common residue was conserved. **C** In silico predicted structure of CASD1 and (**D**) NXPE1 showing co-localization of catalytic site serine from the Gly-Asp-Ser (GDS) motif with catalytic site histidine from the DXXH motif. All structures shown in this figure are predictions based on AlphaFold accessed 11/2023.

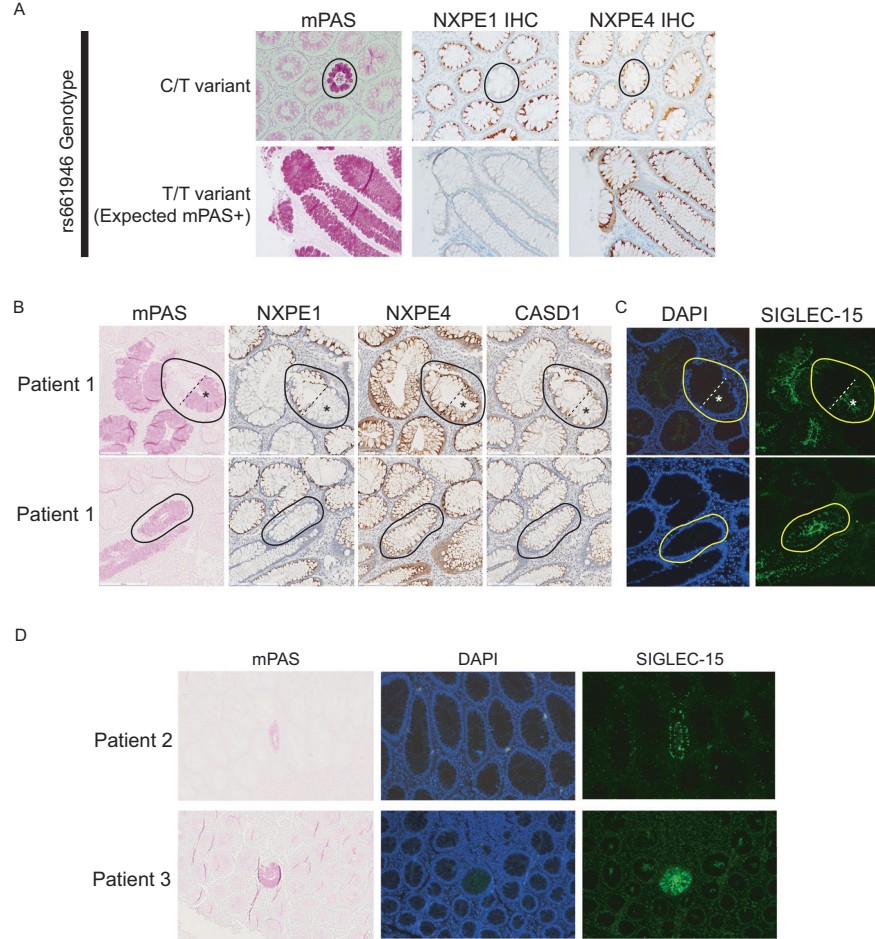

**Fig. 3 | NXPE1 protein expression is correlated with mPAS and SIGLEC-15 staining. A** mPAS, NXPE1 IHC and NXPE4 IHC on adjacent sections from normal colon FFPE tissue with the indicated genotypes for SNP rs661946. Heterozygous samples were primarily negative by mPAS, and positive for NXPE1 and NXPE4, but the images chosen show rare spontaneously mPAS positive crypts. Images are representative examples from more than 20 unique patients, and are shown at 20x. **B** Adjacent sections of normal colon FFPE tissue heterozygous for rs661946 containing spontaneous mPAS positive crypts stained by IHC for NXPE1, NXPE4 or known sialic acid O-acetyltransferase CASD1. Circles highlight the same crypt in each sample. Note that the crypt in patient 1 denoted with a * is positive only in the

bottom half of the crypt. NXPE1 protein appears only in the portion of the crypt that is negative for mPAS. Images are representative examples of 10 patients with similar results and are shown at 20x. **C** Adjacent sections from the same samples in B now stained with SIGLEC-15 and detected by immunofluorescence and DAPI. SIGLEC-15 staining matches the mPAS staining pattern. All images are shown at 20x. **D** Adjacent sections stained with mPAS or immunofluorescence with SIGLEC-15 on normal FFPE colon tissue heterozygous for the *NXPE1* promoter SNP rs661946. mPAS and SIGLEC-15 stain the same cells/crypts. All images are 20x and representative of experiments repeated in at least 10 unique patients.

activity would increase the number of modified sialic acids and lower binding of SIGLEC-15 (Fig. 4A and Supplementary Fig. 6, Supplementary Data 4). To confirm these results, we then performed IHC and flow cytometry using an anti-sialyl-Tn antibody and observed marked reduction in staining for Jurkat cells with NXPE1 overexpression, presumably due to NXPE1-mediated sialyl-Tn O-acetylation (Fig. 4B and Supplementary Fig. 7).

### *NXPE1* displays differential allelic expression

A review of the literature suggests that expression of NXPE1 protein is largely confined to the colon and rectum[41,55,56]. Consistent with this, the RNA expression levels of *NXPE1* were queried in the GTEx database and found to be predominantly expressed in normal human colon tissues with an apparent bimodal pattern in the transverse colon (Supplementary Fig. 8)[41,56]. Furthermore, similar GTEx-based

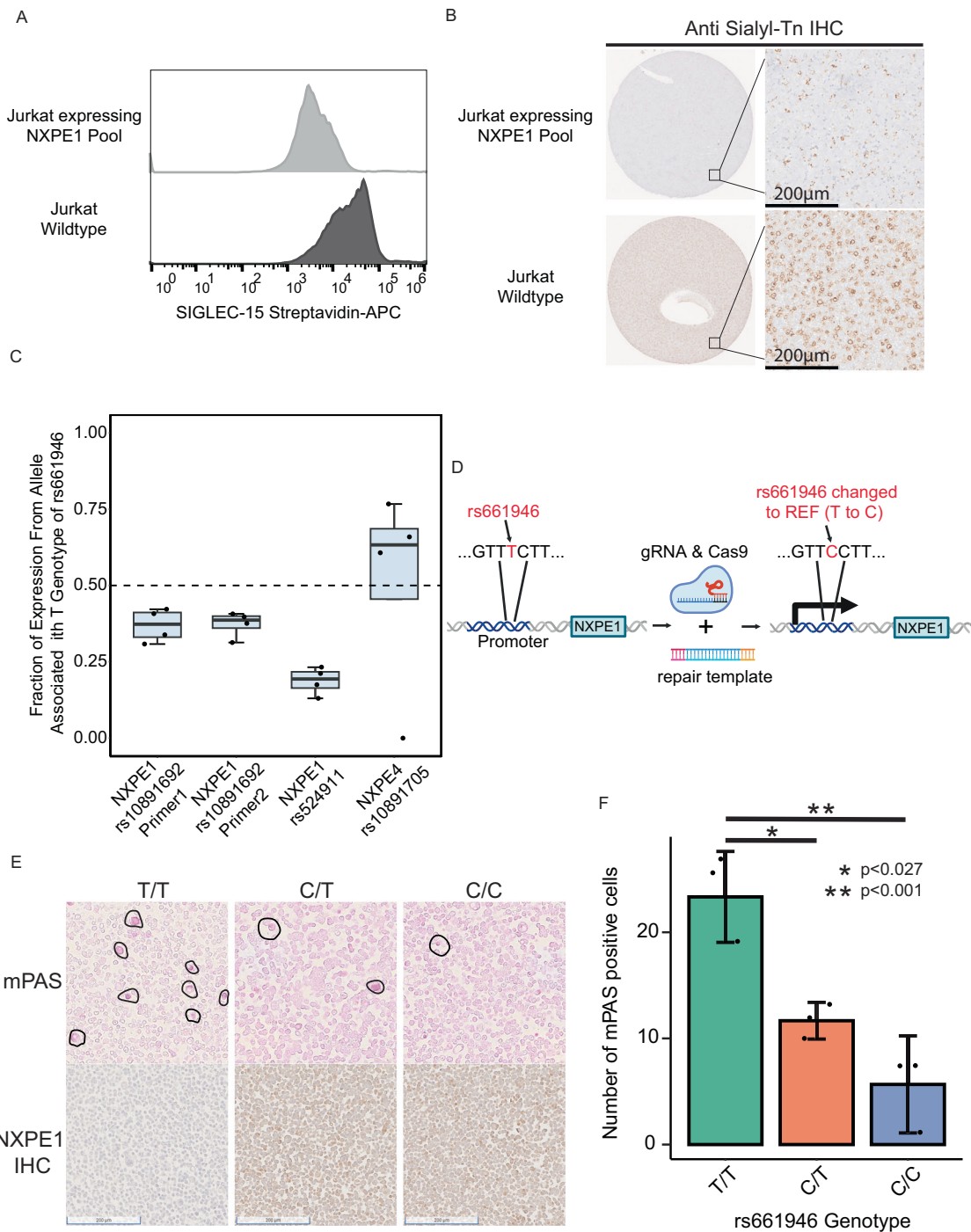

**Fig. 4 | Manipulation of NXPE1 protein expression alters mPAS and SIGLEC-15 staining. A** Flow cytometry with biotinylated SIGLEC-15 conjugated to APC on a pooled population of Jurkat cells infected with lentiviral clones containing a NXPE1 expression cassette. **B** Sialyl-Tn IHC staining on Jurkat wildtype and a pooled population of Jurkat cells overexpressing NXPE1. Images shown are representative of 2 unique pools of cells. Images are shown at 2x and 20x. **C** Fraction of total RNA expression from the allele in linkage with the T allele of rs661946 in normal colon tissue. Data is from targeted deep RNA-sequencing comparing the transcript levels of each allele using coding region heterozygous SNPs in NXPE1 in perfect genetic linkage with the promoter SNP rs661946 in all samples tested. More specifically, the C and A variants of rs524911 and rs10891692, respectively, are tightly linked with the T allele of rs661946, which lacked detectable NXPE1 expression above (Fig. 3A). The coding region SNP from the gene *NXPE4* is shown as a control. Boxplots show

median value, box indicates 75th and 25th quartile and whiskers extend to the farthest value (largest and smallest). $n = 4$ biological replicates for each boxplot. **D** Cartoon describing CRISPR-Cas9 knock in approach to change rs661946 in LS180 cells from T/T (homozygous VAR) to C/T (heterozygous) or C/C (homozygous REF). Red letter indicates base changed for rs661946. **E** mPAS and NXPE1 IHC staining on LS180 cells with the indicated genotypes for rs661946, with heterozygous (C/T) and homozygous REF (C/C) being created by CRISPR-Cas9 knock in. Cells positive for mPAS staining are circled. **F** Counts of the number of mPAS positive LS180 cells with the three possible genotypes for rs661946 as created by knock-in. Quantitation was performed using ImageJ on three different random images from each cell plug at 10x. Error bars indicate 95% confidence intervals, bar tops indicate mean. *P* values are as indicated by one-sided Student T-Test.

transcriptomic analyses revealed that haplotypes in this region were associated with quantitative differences in both transcript levels and altered splicing in a tissue-specific manner (Supplementary Fig. 8 and Supplementary Data 5). In particular, SNPs in tight linkage with the perfectly matched SNP rs661946 noted above in the region displayed effects on *NXPE1* expression in colon (Supplementary Data 5). We were able to independently confirm the allelic effects on expression levels using fresh frozen colon samples. We assessed the RNA expression levels of each allele of *NXPE1* by targeted RNA sequencing of transcript levels in patients heterozygous for coding SNPs in *NXPE1*. Again, the alleles associated with reduced NXPE1 protein expression were associated with a 25–50% reduction in relative transcript levels depending on which exons were queried (Fig. 4C). Finally, it should be noted that *NXPE1* has numerous isoforms and appears to undergo complex tissue specific splicing (Supplementary Fig. 8). This raises the possibility that differences in expression and splicing are regulated by variants in the region accounting for the differences in NXPE1 protein levels. This notion was particularly intriguing given our most closely associated SNP, rs661946, was in the promoter region of one of the dominant colon isoforms (Supplementary Fig. 8).

We next wanted to test the possibility that the rs661946 SNP in a promoter of *NXPE1* is directly contributing to the inherited changes in protein expression noted earlier in this study. To evaluate the effects of changing the T allele to a C allele, colorectal cancer cell lines were surveyed to find one that stained positively with mPAS (Supplementary Fig. 9). We identified one such cell line (LS180), and confirmed the genotype for this cell line as homozygous for the T allele at rs661946, as expected from its mPAS staining (Supplementary Fig. 9, Supplementary Data 4). CRISPR was used to change the T/T genotype in this line to either the heterozygous C/T or homozygous C/C (Fig. 4D and Supplementary Fig. 10). A robust increase in NXPE1 protein and decrease in mPAS staining was observed in the engineered lines with T/T to C/T or C/C change, an observation further supporting the autosomal dominant behavior of *NXPE1* (Fig. 4E, F).

### NXPE1 transfers acetyl to cytidine-5-monophospho-N-acetylneuraminic acid

We wanted to determine if NXPE1, like CASD1, is able to biochemically transfer an acetyl group from acetyl coenzyme A onto cytidine-5-monophospho-N-acetylneuraminic acid (CMP-Neu5Ac), a modified sialic acid used for biosynthesis of sialic acid derivatives[29]. Co-incubation of the predicted extracellular domain of NXPE1 (AA 60-547) with required co-factors and CMP-Neu5Ac led to a modified sialic acid product running at ~11 minutes, suggesting that NXPE1 mediates addition of acetyl groups to sialic acids with the ninth carbon position being most likely (Fig. 5A and Supplementary Fig. 11). Additional minor products at ~7 minutes likely suggest the formation of other acetylated sialic acid derivatives, including at the seventh carbon position (Fig. 5A and Supplementary Fig. 11). To further pinpoint the enzymatic mechanism of NXPE1, we co-incubated NXPE1 with co-factors and CMP-Neu5Ac, but without acetyl coenzyme A. Expectedly, we observed a complete absence of modified sialic acid product, verifying that acetyl coenzyme A is a necessary co-factor for NXPE1's enzymatic function (Fig. 5A and Supplementary Fig. 11). Additionally, we mutated the presumed catalytic serine residue S355 (Fig. 2A & 2D) to an alanine (S355A) in an effort to create a catalytically inactive NXPE1 mutant (NXPE1[S355A]). Encouragingly, we noted complete abrogation of NXPE1-mediated O-acetylation, demonstrating the essentiality of S355 for enzymatic activity (Fig. 5A and Supplementary Fig. 11). Analogous alanine mutation of a spatially proximal aspartic acid residue D526A reduced, but did not completely eliminate, NXPE1 activity (Supplementary Fig. 11). Lastly, we obtained CMP-N-glycolylneuraminic acid (CMP-Neu5Gc), a modified sialic acid that is primarily present in human tissue through dietary exposure, and performed similar biochemical assays with NXPE1[57]. We observed a small amount of modified product

running at ~9 minutes, in alignment with Neu5Gc,9Ac from the sialic acid reference, suggesting that NXPE1 may weakly accept CMP-Neu5Gc as a substrate (Supplementary Fig. 11). Our observations support a model where (1) rs661946 SNP directly modulates the expression of *NXPE1* resulting in variation in NXPE1 activity and (2) NXPE1 mediates sialic acid O-acetylation (Fig. 5B).

## Discussion

The search for genes responsible for O-acetylation of sialic acid has been ongoing for several decades, but despite extensive efforts, often using biochemical approaches, only *CASD1* has been described[3,28]. The approach taken in this study was different than those used previously, and our conclusion that *NXPE1* is the acetyltransferase responsible for mPAS staining is based on the following evidence:

(i) Whole genome sequencing of DNA showed that multiple SNPs on chromosome 11q23 were significantly correlated with mPAS staining in colon tissues. No other chromosomal positions were perfectly linked to this phenotype (Fig. 1A and B).

(ii) More refined genomic and histochemical studies showed that only a single variant - rs661946, 6 bp upstream of a transcriptional start site of *NXPE1* - was perfectly correlated with mPAS staining in all 112 patients studied (Fig. 1D and Supplementary Data 3).

(iii) *NXPE1* contains motifs identical to those in the catalytic site of the only know human sialic acid-O-acetyltransferase (SOAT), as well as viral homologs of SOAT (Fig. 2).

(iv) Immunohistochemical studies with two different antibodies to NXPE1 showed perfect correlations with mPAS staining in the colon. In particular, the rare crypts that were mPAS positive in rs661946 C/T heterozygotes did not express detectable levels of NXPE1 and were immersed in a sea of crypts that did express NXPE1 (Fig. 3A).

(v) Of twenty sialic acid-binding proteins in the SIGLEC family tested by immunofluorescence, only SIGLEC-15 consistently stained the same crypts that were stained by mPAS, both in different patients and in different crypts of the same patient. In particular, the rare crypts in rs661946 C/T heterozygotes that bound to SIGLEC-15 were found to be mPAS positive (Fig. 3C, D).

(vi) Overexpression of NXPE1 in cell lines resulted in loss of SIGLEC-15 and sialyl-Tn binding (Fig. 4A, B).

(vii) In patients heterozygous for the promoter SNP rs661946 and the transcribed SNPs rs10891692 and rs524911, *NXPE1* transcripts from the alleles containing T were reduced compared to those containing C. GTEx data confirms that there is an allelic imbalance of expression for SNPs located near the *NXPE1* promoter (Fig. 4C).

(viii) Knock-in of a C in place of a T allele for rs661946 in a colon cell line resulted in the expected increase in NXPE1 protein and corresponding decrease in mPAS staining (Fig. 4E, F).

(ix) NXPE1 protein can acetylate sialic acids in vitro (Fig. 5A).

While these lines of evidence above strongly implicate *NXPE1* as the SOAT responsible for mPAS staining, many questions remain unanswered. Based on sequence alignments and predicted 3D structures, the catalytic site for *NXPE1* is similar to that of *CASD1*. Indeed, mutating the putative catalytic serine at position 355 to an alanine (S355A) and spatially proximal aspartate at position 526 to an alanine (D526A) abrogated NXPE1 function. The specifics of how the acetyl-transfer is performed, the location on sialic acid targeted by NXPE1, and the potential relationship between CASD1 and NXPE1 remains to be investigated. The preference for SIGLEC-15 to bind NXPE1 unmodified proteins suggests that these protein targets contain Ser/Thr-GalNac linked sialic acid at the α2-3 or α2-6 position[50,54,58]. However, this remains speculative as we've only demonstrated that enzymatically active NXPE1 is capable of sialic acid O-acetylation, but not the

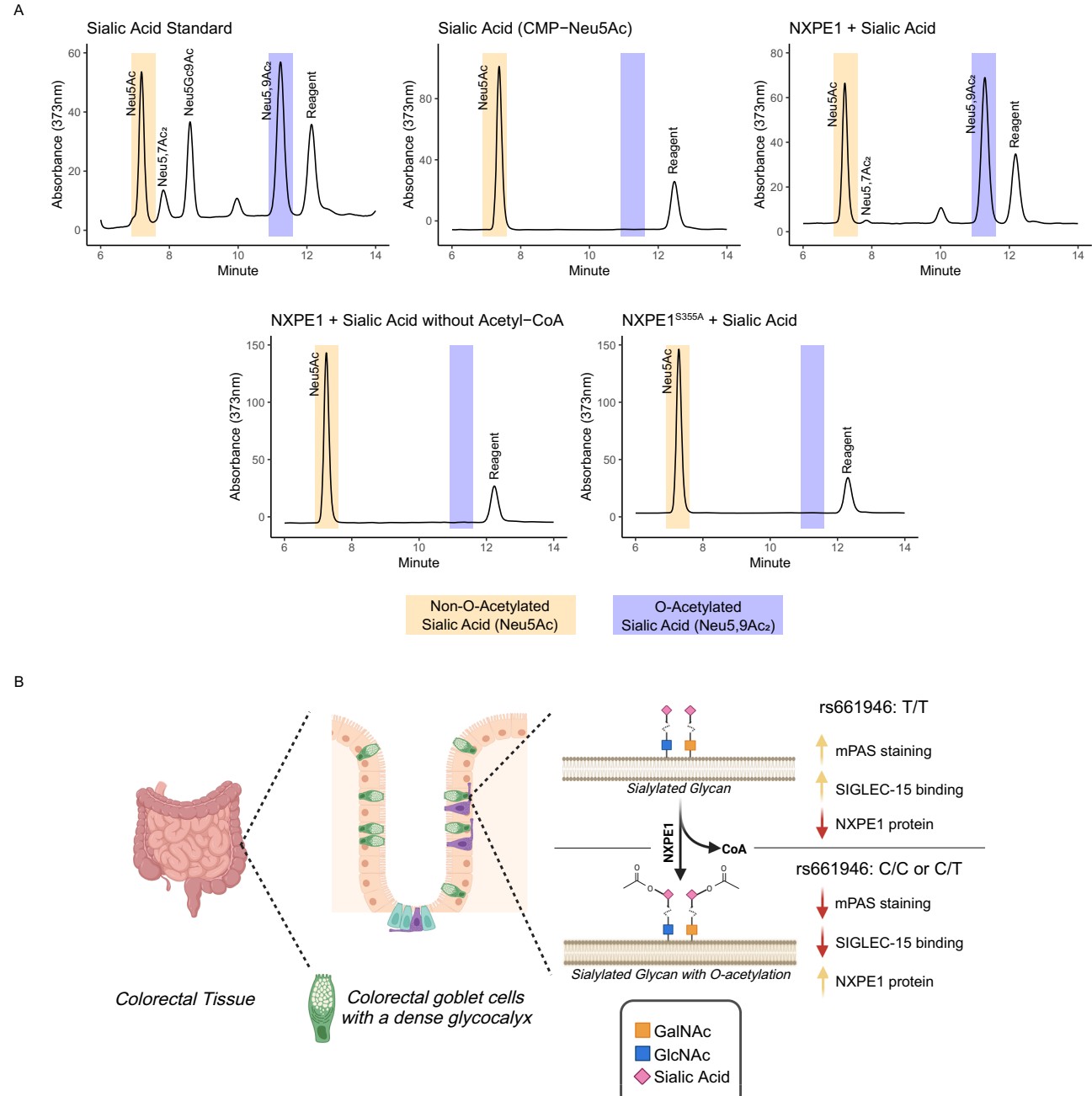

**Fig. 5 | NXPE1 mediates O-acetylation of sialic acids in vitro. A** DMB-based HPLC readout of an enzymatic reaction containing CMP-Neu5Ac and acetyl coenzyme A, with or without the predicted extracellular domain of NXPE1 (AA 60-547) or a catalytically inactive mutant NXPE1$^{S355A}$. Note that the Sialic Acid Standard represents a panel of sialic acid derivatives that were identified per manufacturer's instructions. Non-O-acetylated sialic acid (yellow shading) appears at about 8 min and acetylated sialic acid (blue shading) appears around 12 min. **B** Cartoon summarizing proposed NXPE1 activity in colorectal tissue. Note that the targeted sialylated glycans shown, GalNAc and GlcNAc, are based on previously published data describing SIGLEC-15's preferred binding partners. Goblet cells are shown in green, and enteroendocrine cells in purple. Created in BioRender. Paul, S. (2025) https://BioRender.com/zjv09id.

orientation of the sialic acid with the peptide backbone. Furthermore, the *NXPE* family has four members, none of which have been thoroughly characterized, and the potential role of the other three members in modifying the sialoglycome remains speculative. In silico 3D structures, however, predict that all of the family members contain a conserved GDS and HPP motif exposed to solvent (Supplementary Fig. 3). This leads us to speculate whether they may also perform similar sialic acid acetylation activity in other tissues, on different positions of sialic acid, or on sialic acids with different linkages[36].

During the revision of our study, Humeidi and colleagues reported encouraging results on the role of NXPE1 as a sialic acid

O-acetyltransferase in their preprint[59]. Their study corroborated our findings that NXPE1 is capable of acetylating sialic acids at the ninth carbon position and that enzymatic activity is largely dependent on the catalytically active serine at position 355. Curiously, they identified a missense variant (G353R) encoded by the transcribed SNP rs10891692 which seemed critical for protein folding in two exogenous expression systems. Of note, this SNP was identified in our WGS and targeted sequencing studies, wherein it perfectly correlated with all but one sample (111/112 samples) (Fig. 1D and Supplementary Data 3). Provided its close proximity (< 30 kb) and tight linkage with the promoter SNP

**Table 1 | Primer sequences for targeted sequencing**

| Forward Seq | Reverse Seq | SNP |
|---|---|---|
| AAAAGCAGCCTCGAGACATT | TCACGAGAAACCCACTCCCA | rs34521319 |
| CCTACCAAACATGTTAGCCT | TGCTGTTCAGTGTCTTGGCGA | rs545355 |
| AGCCACACAATTAATAACTACCA | CTCAGTCTCTCCACCAACCCT | rs561722 |
| GTGGCTGGAAAAGTACCTGTGA | TGCCTCTTCAAATCTCCACCCC | rs661946 |

rs661946, and it changing the functionally relevant glycine in the GDS active site of NXPE1, we cannot rule out the possibility that rs10891692 may also contribute to the expression of functional *NXPE1*. However, we should note that knocking in the rs661946 allele supporting expression was sufficient to restore detectable NXPE1 protein and activity in LS180 which is homozygous for the rs10891692 allele reported to be unstable in their exogenous expression systems.

There are potential practical implications of our findings. Genome-wide association studies have shown that the genomic region containing *NXPE1* is linked to inflammatory bowel disease, as well as a non-significant trend for metastatic colorectal cancer risk[42,60,61]. It is possible that *NXPE1*, or the closely positioned *NXPE4*, are responsible for this linkage[62]. If so, then one could imagine that NXPE1-targeted therapies could be used to treat a subset of patients with this disease if *NXPE1* is confirmed as a contributing factor. Second, loss of 11q23 is found in more than 20% of colorectal cancers[63,64]. Across the general population with an allele frequency of 0.38, ~50% of these patients will be C/T heterozygotes at rs661946. When the C allele is lost in the cancer cells of these patients, they are no longer O-acetylated on the sialic acids recognized by mPAS or SIGLEC 15. This creates potential opportunities for specifically targeting the non-O-acetylated cancer cells while sparing healthy O-acetylated cells. With potential to impact health and wellbeing in a number of ways, there is clearly a lot left to explore in this exciting field.

## Methods

This study was conducted in accordance with all relevant ethical regulations established at Johns Hopkins University School of Medicine. Patient samples for this study were all commercially sourced, and are available to the public for use. As such, no specific committee or body was required to approve use of these samples. Information on samples was stored and access limited to necessary users only as required by Johns Hopkins University use of human samples. Sex/gender was not considered during the study design, and sex was either not supplied or determined based on reporting by the commercial company. There are no published data to date supporting that sex impacts the phenotype, mPAS staining, studied in this manuscript.

### Mammalian cell lines and cell culture

Jurkat, HEK293T, and LS180 cells were purchased from The American Type Culture Collection (Virginia, USA). Jurkat cells (ATCC, Cat #TIB-152) are male in origin and were grown in RPMI 1640 Medium (ATCC, Cat #30-2001), supplemented with 10% fetal bovine serum (HyClone, Utah, USA, Cat #16777-006) and 1% Penicillin-Streptomycin (Gibco, USA, Cat #15140122). LS180 cells (ATCC, Cat #CL-187) are female in origin and were grown in EMEM (ATCC, Cat #30-2003), supplemented with 10% FBS (HyClone, Utah, USA, Cat #16777-006) and 1% Penicillin-Streptomycin (Gibco, USA, Cat #15140122). HEK-293T cells (ATCC, #CRL-3216) were grown in DMEM (Gibco, USA # 11965092) supplemented with 10% FBS (HyClone, Utah, USA, Cat #16777-006) and 1% Penicillin-Streptomycin (Gibco, USA, Cat #15140122). In vitro cells were maintained at 37°C with 5% CO2. Mycoplasma testing was performed by The Genetic Resources Core Facility at Johns Hopkins School of Medicine (Maryland, USA).

### Whole genome sequencing (WGS) workflow

21 samples: 8 mPAS positive, 8 mPAS negative, and 5 mPAS negative with rare positive crypts - were selected for whole genome sequencing. Investigators were not blinded during analysis. Matched FFPE sections and plasma pellets were commercially obtained from ILSbio LLC (Chestertown MD) for each sample. DNA was purified from on average 5 mL of plasma using a DNeasy Blood & Tissue Kit (Qiagen 69504), and quantified by Nanodrop 2000 (ThermoFisher Scientific). Library preparation was performed using a ThruPLEX® DNA-Seq kit (Takara R400675) per manufacturer's instructions. Each sample was sequenced on 3 or 4 lanes of a Hiseq 4000 (Illumina) with paired end 100 cycle settings per manufacturer's instructions. Fastq files for each sample are publicly available at the European Genome-Phenome Archive (https://ega-archive.org/) under accession number EGAS00001007704. VCF files were created for each using the cancer genomics cloud (Seven Bridges)[65]. Briefly, alignment and VCF generation was performed with Whole Genome Analysis - BWA + GATK 2.3.9-Lite[66,67]. VCF files were then moved into SQL for further analysis including genotype assignment and comparison. SNPs were considered a perfect genotype-phenotype match if one of two conditions were met: 1) all positive mPAS staining samples contained an alternate (VAR) genotype, negative staining with rare positive contained a heterozygous genotype, and negative staining contained either a heterozygous or homozygous reference genotype. 2) All mPAS positive staining samples contained a reference genotype, negative mPAS staining with rare positive contained a heterozygous genotype, and negative mPAS staining samples contained a heterozygous or homozygous alternate (VAR) genotype. Data was also analyzed in a less stringent form, allowing for 1–2 mismatches between genotype and phenotype. Variant consequence analysis was performed using the Variant Effect Scoring Tool (https://www.cravat.us/CRAVAT/)[68]. Plot of perfectly concordant SNPs (Supplementary Fig. 1) was performed using the package mapsnp in R[69]. GWAS and association case-control analysis was performed on 10,421,023 variants using PLINK v2.00a6 64-bit with the following settings: gender set as a covariate, eliminated SNPs with a minor allele frequency below 0.1, assumed dominant genotype, only considered biallelic SNPs, and removed SNPs not in Hardy Weinberg Equilibrium. Linkage disequilibrium (LD) and haplotype block analysis was performed using PLINK v1.90b7.2 64-bit (11 Dec 2023), pairwise LD comparisons were made between all SNPs within 1 Mb of each other on chromosome 8 and 11. GWAS study plots generated with the R packages qqman v0.1.9 (http://pngu.mgh.harvard.edu/purcell/plink/)[70,71], GWLD v1.3.6[72] and Ldheatmap v1.0-5.

### Targeted sequencing workflow

Targeted sequencing validation was performed on DNA from normal colon or normal adjacent tumor FFPE samples obtained commercially from ILSbio LLC (Chestertown, MD). Investigators were not blinded during this experiment. Six 5 µm sections were pooled together into a single tube for each of the 91 samples. DNA was extracted using a QIAamp DNA FFPE Tissue Kit (Qiagen 56404). DNA was quantitated using Nanodrop 2000 (ThermoFisher Scientific). PCR was performed on target locations using Phusion Flash High-Fidelity PCR Master Mix (ThermoFisher Scientific F548L) following manufacturer's instruction and primers listed below (Table 1). Sanger sequencing of the product was performed by Genewiz.

## Multiple sequence alignment and in-silico 3D structure prediction

Multiple sequence alignment was done using NCBI-BLASTp, COBALT and Jalview[73–75]. 3D structure prediction was accomplished using AlphaFold[76]. All were accessed 11/2023.

## Overexpression in cell lines

Overexpression cassettes (pLenti-C-mGFP-P2A-Puro Vector #RC206474L4 from Origene) were transfected into HEK-293T cells with MegaTran 2.0 (Origene TT210002) with psPAX2 (Addgene #12260) and pMD2.G (Addgene #12259) in a 10:3:1 ratio (30 μg of total plasmid DNA per well) respectively following manufacturer's instructions. Lentivirus was harvested after 48 hrs, and a second time at 72 hrs. Harvests were combined, run through a 0.45 PES filter (Millipore-SLHPM33RS), and concentrated with Lenti-X concentrator (Takara #631232). Virus stocks were stored in 1 mL of DMEM (Gibco, USA # 11965092) with 10% FBS (HyClone, USA, Cat #16777-006) and concentrations determined by p24 ELISA (Takara #632200). Nontreated 48 well plates were treated with RetroNectin, according to manufacturer's instructions (Takara #T100). Using RetroNectin-treated plates, 50,000 target cells were transduced by spinoculation at $800 \times g$ for 2 hrs at 37°C with 300 μL virus. Cells were washed and then monitored for GFP and target gene expression. Selection for 14 days was performed with 1 μg/mL of puromycin (InvivoGen ant-pr-1).

## Allele expression on colon tissue

Fresh frozen normal colon tissue was obtained from BioIVT (Westbury, NY). Investigators were not blinded during these experiments. RNA was purified from on average 10–20 mg of tissue using a RNeasy Mini Kit (Qiagen 74104) following manufacturer's instructions, and quantified by 4200 Tapestation (Agilent). RNA was converted to cDNA using a High-Capacity cDNA Reverse Transcription Kit (ThermoFisher Scientific #4368814) following manufacturer's instructions. PCR was performed using primers listed below (Table 2) and Phusion Flash High-Fidelity PCR Master Mix (ThermoFisher #Scientific F548L) following manufacturer's instruction for 35 cycles. PCR products were then purified with AMPure beads (Beckman Colter, California, USA, Cat #A63880) and run on a second round (2–6 cycles) of PCR as previously described[77] to add barcodes. Libraries were pooled, cleaned up with AMPure beads (Beckman Colter, California, USA, Cat #A63880) and sequenced on an Illumina Miseq using manufacturer's instructions (150 cycle single read). Allele fractions were determined by processing fastq files using HISAT2 (version 2.0.5) and aligning to a pseudo reference genome consisting of only the mutant or wild-type amplicon sequences for targeted regions. The allele fraction was determined by taking a ratio of the number of transcripts from the alternative allele to the total number of transcripts from the region in question. Initial data processing was performed in MSSQL and Excel.

## CRISPR KI of cell lines

gRNA was obtained from Integrated DNA Technologies (IDT) (#CD.HC9.KDJS8530.AA), repair template was also obtained from IDT (#CD.HC9.QPVN2588) (Table 3). gRNA complex was performed by mixing gRNA (50 μM final concentration) and Alt-R® CRISPR-Cas9 tracrRNA (IDT #1072533)(50 μM final concentration), heating to 95°C for 5 min then allowing to cool to room temperature on benchtop. RNP complex was performed by mixing 3 μL of the above gRNA complex with 2.4 μL of Alt-R™ S.p. Cas9-GFP V3 (IDT #10008161) (final concentration 30 μM for each) and incubating at room temperature for 15 min. LS180 cells were trypsinized (Gibco # 25200056), pelleted and washed twice with PBS (Gibco # 10010023), then suspended in SG Nucleofection buffer (Lonza, #V4XC-3024) at 25e6cells/mL. A total of 5e5 cells were electroporated by mixing 5 μL of RNP complex from above, 1.2 μL of HDR Donor Oligo, 1.2 μL of Alt-R™ HDR Enhancer V2

(IDT #10007910), 20 μL of the LS180 cell suspension in SG buffer (see above), and 0.1 μL of PBS (all reagents final concentration of 5 μM). Cells were immediately electroporated on a 4D Nucleofector X-Unit (Lonza, AAF-1003X) using pulse code DS-150A. Cells were then removed and plated into complete media and incubated for 5 days. Once enough cells were present, cells were clonally selected by limited dilution as previously described[78]. Accurate knock in was confirmed by PCR using Phusion Flash High-Fidelity PCR Master Mix (ThermoFisher #Scientific F548L) following manufacturer's instruction and primers listed below. Sanger sequencing of the product was performed by Genewiz.

## Immunohistochemistry (IHC) analysis

Immunostaining was performed at the Oncology Tissue Services Core of Johns Hopkins University School of Medicine. Immunolabeling was performed on formalin-fixed, paraffin embedded sections on a Ventana Discovery Ultra autostainer (Roche Diagnostics). Briefly, following dewaxing and rehydration on board, epitope retrieval was performed using Ventana Ultra CC1 buffer (Roche Diagnostics, Cat #6414575001,) at 96°C for 64 minutes. Primary antibody, anti-CASD1 (1:100 dilution; Invitrogen, Cat #PA5-60700, Lot #Xi3700359), anti-NXPE4 (1:1000 dilution; Sigma-Aldrich, Cat #HPA042801, Lot #R39941), anti-NXPE1 (1:200 dilution; Santa Cruz Biotechnology, Cat #sc-514349 and 1:100 dilution; catalog# HPA049133, Lot# R59143, Sigma-Aldrich) or Anti-Sialyl Tn antibody [STn 219] (1:50 dilution; Cat #ab115957, Lot 1063860-1, Abcam) diluted in Antibody Diluent with Casein (Roche Diagnostics, Cat #6440002001,); was applied at 36°C for 60 minutes. Primary antibodies were detected using an anti-rabbit HQ detection system (Roche Diagnostics, Cat #7017936001 and 7017812001,) followed by Chromomap DAB IHC detection kit (Roche Diagnostics, Cat #5266645001), counterstaining with Mayer's hematoxylin, dehydration and mounting. Note that figures shown in this manuscript for NXPE1 IHC were performed with primary antibody from Santa Cruz, though both the Sigma and Santa Cruz antibodies provided similar results.

## Mild periodic acid Schiff staining

Staining for mPAS was performed by the Johns Hopkins Reference Pathology Lab. Briefly, slides were dewaxed and hydrated in distilled water. Slides were then washed in 0.1 M acetate buffer, pH 5.5 (Thermo Fisher, Cat #AM9740), at 2°C for five minutes. Slides were subsequently treated with 1 mM (0.02%) NaIO$_4$ (Sigma Aldrich, Cat #P7875-100G) in 0.1 M acetate buffer, pH 5.5, at 4°C for 2 min, and then washed in 1 % aqueous glycerol (Sigma Aldrich, Cat #G5516) for five minutes. Slides were washed in distilled water for five minutes and treated with Schiff's reagent (Sigma Aldrich, Cat #3952016-500 ML) at room temperature for 15 minutes. Slides were washed three times in 0.5% K$_2$S$_2$O$_5$ (Sigma Aldrich, Cat #60508) in 0.05 M hydrochloric acid (Sigma Aldrich, Cat #2104-50 ML) for five minutes, and then washed in running tap water for five minutes. Slides were then washed in distilled water for five minutes. Finally, slides were dehydrated, cleared, and mounted with a coverslip.

## Immunofluorescence (IF) analysis

Blocking was performed by incubating paraffin-embedded tissue samples using 1% BSA with 2% FBS in PBS-T, for a total of 30 minutes. Immunostaining was performed by incubating samples with both primary and secondary antibodies overnight at 4°C using the following reagents: recombinant human Siglec-15-Fc (5 μg/mL; R&D Systems, USA, Cat #9227-SL-050) and Anti-Human IgG Alexa 488 (3 μg/mL; Jackson ImmunoResearch, USA, Cat # 109-545-170). Samples were thoroughly washed with PBS and mounted in Prolong Gold Antifade Reagent with 4', 6-diamidino-2-phenylindole, dihydrochloride (DAPI). Other SIGLEC constructs tested were also obtained from R&D Systems and used at an equivalent concentration.

## Table 2 | Primers sequences used for allele specific expression analysis

| SNP* | Forward Validation Primer | Reverse Validation Primer |
|---|---|---|
| rs10891692 | CGACGTAAAACGACGGCCAGTNNNNNNNNNNNNNACATTTTGCAACCAGGTTCAGT | CACACAGGAAACAGCTATG**ACCATGAGATTCCAGTTTCATGAAGATCAAAA** |
| rs524911 | CGACGTAAAACGACGGCCAGTNNNNNNNNNNNNNNCAGTCTCTGTTAGTGGCTTT | CACACAGGAAACAGCTATG**ACCATGGCTTGGTCTGCTCTAAACT** |
| rs10891705 | CGACGTAAAACGACGGCCAGTNNNNNNNNNNNNNNCCATAGTGTCTGGGCTTCTCA | CACACAGGAAACAGCTATG**ACCATGTCCAAATGCAACAAGAAACAGT** |

*Note validation primers include M13F or M13R universal primer sequence and a 13 bp unique identifier (UID) concatenated to them for use in attaching barcoding after specific target amplification as previously described[7]. Sequence specific for the SNP site is shown in bold.

## Cell staining and flow cytometry

Jurkat cells with NXPE1 overexpression were suspended at $1 \times 10^6$ cells/mL in staining buffer and incubated with constructs at relevant concentrations for 30 minutes on ice, in the dark. Primary staining was performed with SIGLEC-15 (Acro Biosystems, Cat #SG5-H82E9) monomer at a concentration of 5 μg/mL. In our hands, using fresh reagent was preferable for proper staining. Secondary staining was performed with APC-conjugated streptavidin (BioLegend, Cat #405207) at a concentration of 2 μg/mL. For anti-Sialyl Tn antibody staining, primary staining was performed with anti-Sialyl Tn antibody (Abcam, Cat #ab115957) at a concentration of 2 μg/mL followed by an anti-mouse IgG-Alexa Fluor 647 secondary antibody (Cell Signaling, Cat #4410) at a concentration of 10 μg/mL. Single, live cells were isolated using forward and side scatter characteristics. Flow cytometry data was analyzed using FlowJo v. 10.1 software.

## NXPE1 expression plasmid and protein production

Expression construct and protein expression was performed by GeneArt (ThermoFisher Scientific). Briefly, expression constructs for the predicted extracellular domain of NXPE1 (AA 60-547) or mutant variants of NXPE1 (NXPE1$^{S355A}$ and NXPE1$^{D526A}$) were synthesized as human codon-optimized gene fragments and cloned into a pcDNA3.4-TOPO vector backbone with an IL2 secretion signal followed by a 6x histidine tag and GGGG linker, then NXPE1 protein sequence. NXPE1 expression plasmid DNA was purified from transformed DH10B E.coli competent cells through Thermo Fisher Scientific's GeneArt protein expression service. The plasmid created above is stored and available at Addgene (Watertown, MA) under accession 226841. Recombinant NXPE1 was expressed by GeneArt (ThermoFisher) in Expi293 cells and purified using a HisTrap FF 5 ml (Cytiva), and the following buffers: wash buffer (20 mM imidazole, 500 mM NaCl, 25 mM Tris HCl pH 7.4), and elution buffer (500 mM imidazole, 500 mM NaCl, 25 mM Tris HCl pH 7.4). Recombinant NXPE1 was stored at −80 °C in PBS until use.

## In vitro NXPE1 biochemical assay using DMB-based HPLC

Enzymatic assays to determine sialic acid O-acetyltransferase activity in vitro were adapted from[29]. 50 mM MES pH 6.5 (Thermo Scientific Chemicals, Cat #J61587.AK), 10 mM MnCl$_2$ (Sigma-Aldrich, Cat #M1787), 10 mM acetyl-CoA (Sigma-Aldrich, Cat #A2056), and 1.25 mM CMP-Neu5Ac (Sigma-Aldrich, Cat #5.05223) or 1.25 mM CMP-Neu5Gc (Chemily Glycoscience, Cat #SN02020) were incubated with or without 5 μg recombinant NXPE1 (AA 60-547) or its mutant variants (S355A and D526A) in a 20 μL reaction volume for 3 h at 37°C. Sialic acids were then released via acid hydrolysis by adding 2 M propionic acid (Sigma-Aldrich, Cat #402907) to each sample for 1 h at 80°C. Released sialic acid samples were subsequently labeled with 4,5-methylenedioxy-1,2-phenylenediamine dihydrochloride (DMB) following manufacturer's instructions (Agilent, Cat #GKK-407). DMB-labeled samples, reference panel, Neu5Gc standard (Ludger/QA-Bio, Cat #CM-NEU-GC-01), and Neu5,9Ac$_2$ standard (Ludger/QA-Bio, Cat #CM-NEU5,9AC2-01) were analyzed on a GlykoSep R HPLC column (Agilent) via an Agilent 1260 Infinity HPLC system fitted with a 1260 Infinity Multiple Wavelength Detector. Isocratic elution with acetonitrile/methanol/water (9:7:84, v/v) was performed at a flow rate of 0.7 mL/min and absorbance was measured at the excitation wavelength of DMB (373 nm).

## Statistics

Statistics were performed using R version 4.3.2 and RStudio (RStudio 2023.09.0 + 463 "Desert Sunflower" Release (b51c81cc303d4b52b010767e5b30438beb904641, 2023-09-25). GWAS and association case-control $p$ value analysis was performed using PLINK v2.00a6 64-bit, and linkage disequilibrium $r^2$ values obtained using PLINK v1.90b7.2 64-bit (11 Dec 2023) (http://pngu.mgh.harvard.edu/purcell/plink/)[70]. Hardy-Weinberg equilibrium by Haldane's exact test performed using R-package HardyWeinberg (version

**Table 3 | Sequences of oligos used to create and confirm rs661946 knock-in cell lines**

| gRNA sequence | CCAAGATAAGCCAATTTACG |
|---|---|
| Repair template | TTCTTAACTGGCTCTTGTGGCTGTGTAAAATGTTTTTGTTCCTTGTCTGTGAACA-CATTTATTTTCATCAGTCTGTTTTCCTCGTAAATTGGCTTATCTTGGGGTGGAGATTTGAAGAGGCAATA |
| Forward confirmation primer | ACCATGAGCCAGATACCTGATCATTCAGTGA |
| Reverse confirmation primer | TGTGCTCCTCAGTCTCTCCACCA |

1.7.5)[79]. Fisher's Exact Test and student T-test were performed using base R-package stats. Multiple sequence alignment expect values were calculated by the BLASTp software and listed here as reported[75]. pLDDT values were calculated by AlphaFold during modeling and listed as reported. Quantification of knock-in mPAS stained slides was performed using ImageJ and student T-test in Excel[80].

### Reporting summary

Further information on research design is available in the Nature Portfolio Reporting Summary linked to this article.

## Data availability

The Fastq files for all 21 samples that underwent whole genome sequencing generated in this study have been deposited in the European Genome-Phenome Archive [https://ega-archive.org/] database under accession number EGAS00001007704.

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

## Acknowledgements

The authors would like to thank Dr. Sujayita Roy, Sarah Hughes, and Dr. Alan Meeker of the Johns Hopkins Sidney Kimmel Comprehensive Cancer Center Oncology Tissue Services Core for their assistance with IHC staining and optimization. They would also like to thank Dr. Tatianna Larman and Dr. Maged Zeineldin for their help with reviewing the manuscript. The Genotype-Tissue Expression (GTEx) Project was supported by the Common Fund of the Office of the Director of the National Institutes of Health, and by NCI, NHGRI, NHLBI, NIDA, NIMH, and NINDS. The data used for the analyzes described in this manuscript were accessed from the GTEx Portal on 11/2023. Tissue expression information was also accessed from the Human Protein Atlas proteinatlas.org on 11/2023. This work was supported by Oncology Core CA 06973 (B.V., K.W.K., N.P.); The Virginia and D.K. Ludwig Fund for Cancer Research (B.V., K.W.K., N.P., C. B.); The Sol Goldman Sequencing Facility at Johns Hopkins (B.V.); NIH Cancer Center Support Grant P30 CA006973; Lustgarten Foundation for Pancreatic Cancer Research (B.V., N.P., K.W.K., and S.Z.); The Commonwealth Fund (B.V., N.P., K.W.K., S.Z., and C.B.); Howard Hughes Medical Institute (B.V.).

## Author contributions

N.W., K.K., and B.V. conceptualized the study. S.S., S.Z., B.V., N.P., C.B., K.G., P.B., and S.K. provided materials and advice on study and experiment design. N.W., B.S.L., A.C., L.D., M.P., B.V., and K.K. performed experiments. N.W., K.K., and B.S.L. wrote the manuscript.

## Competing interests

B.V., K.W.K., & N.P. are founders of Thrive Earlier Detection, an Exact Sciences Company. K.W.K., N.P. are consultants to Thrive Earlier Detection. K.W.K., N.P., and S.Z. hold equity in Exact Sciences. B.V., K.W.K., N.P., S.S., and S.Z. are founders of or consultants to and own equity in CLASP, Haystack Oncology, Neophore, and CAGE Pharma. N.P. is consultant to Vidium. B.V. is a consultant to and holds equity in Catalio Capital Management. S.Z. has a research agreement with BioMed Valley Discoveries, Inc. C.B. is a consultant to Depuy-Synthes, Bionaut Labs, Haystack Oncology, and Privo Technologies. C.B. is a co-founder of OrisDx and Belay Diagnostics. The companies named above, as well as other companies, have licensed previously described technologies peripherally-related work described in this paper from Johns Hopkins University. B.V., K.W.K., N.P., and S.S. are inventors on some of these technologies. Licenses to these technologies are or will be associated with equity or royalty payments to the inventors as well as to Johns Hopkins University. The terms of all these arrangements are being managed by Johns Hopkins University in accordance with its conflict-of-interest policies. The remaining authors declare no competing interests.
