## [Transparent Peer Review file · Nature Communications]

NXPE1 alters the sialoglycome by acetylating sialic acids in the human colon

Corresponding Author: Dr Nicolas Wyhs

Version 0:

Reviewer comments:

Reviewer #1

(Remarks to the Author)

Lee et al.

NXPE1 alters the sialoglycome by acetylating 2 sialic acids in the human colon.

This is an elegant, complete, and convincing study that solves a long standing open biological question of the basis of acetylation of sialic residues on mucins and other glycoproteins, particularly in the gut. The complementary utilization of population genetics, molecular biology, structural biology, and biochemical techniques convincingly establishes that NXPE1 is responsible for acetylation of sialic acid residues, and strongly suggests similar activity will be demonstrated by other members of the gene family. As pointed out in the discussion, there are several significant translational implications of the finding for identifying new therapeutic targets for IBD and colorectal cancer.

Some minor comments are:

1. It would perhaps merit a comment as to what the LD between rs661646 and the SNPs assessed in the QTL analysis, and the effect size seen in the QTL model (-0.2 per allele in a linear model of the transverse colon), suggest as to the effect size of the rs661646 alleles.
2. It would be of interest to show loss of mPAS staining on NXPE1 transfected Jurkat cells, or to comment on why this was not assessed. Given the completeness of the study, the reviewer imagines there was some technical issue with the mPAS assay on these cells.
3. It may be of interest to show an alignment or spatial model that displays the glycine and asparagine residues that are noted as lacking in NXPE1, but as being characteristic of SOATs from other species.
4. It may prevent misunderstanding to reword:
"The preference for SIGLEC-15 to bind NXPE1 modified proteins suggests that these protein targets contain Ser/Thr-GalNac linked sialic acid at the α 2-3 or α 2-6 253 position." to clarify that SIGLEC-15 binds to the same proteins that can be modified by NXPE1 (as SIGLEC-15 binding is actually blocked from binding to proteins that have been modified/acetylated by NXPE1).

Reviewer #2

(Remarks to the Author)

The authors present an interesting multidisciplinary study that suggests the identify a putative sialic acid O-acetyltransferase (SOAT) NXPE1 in the colon. The authors utilise a well-established mild periodic acid Schiff staining (mPAS) that can distinguish, in part, between O-acetylated (mPAS negative) and non-O-acetylated (mPAS positive) sialic acids in tissues.

To the best of my knowledge, the only characterised human sialic acid O-acetyltransferase (SOAT) is CASD1, and consequently the present study may bring to light new information in a very important area of sialoglycobiology and cancer.

Some General comments: (1) PAS is not selective - so the cells could be filled up with glycogen and they would still turn red. Was the staining done with a preliminary distase treatment? Glycogen metabolism is known to be altered in colon cancer too. A glycomic profile of the cells allegedly expressing Neu5,9Ac2 would add significant strength to the study and in my view is more reliable than PAS.

(2) The authors refer to NXPE1 as 'extracellular' - implying CMP-Neu5Ac and AcetylCoA should be found in the extracellular environment in order to work - but then it is stated that NXPE1 is in the intra-endoplasmic reticulum/golgi space. Further histo for colocalization studies are required to better characterise the enzyme's location.

(3) Chromatograms - ABSORBANCE was measured at 373 nm (?) There is an evident 'bump' following the Neu5Ac peak that appears to correspond to Neu5,7Ac2 and so perhaps NXPE1 produces this product or both it and Neu5,9Ac2. Further functional characterisation is required and I would recommend evaluation of deoxy-Neu5Ac analogues to assess what the enzyme really does and/or spend some time to try to elucidate the real mechanism?

(4) The observation that this enzyme doesn't accept Neu5Gc as a sialic acid could be true - but it's puzzling considering colon cancer does express Neu5Gc - aka did they feed Neu5Gc to the enzyme? How did the authors prepare CMP-Neu5Gc..? It is stated that "N-glycolylneuraminic acid (Neu5Gc) is a sialic acid not found in humans and it's used as a control". Which is at least partially wrong, as we know that it is found BUT not synthesised.

(5) There is no information regarding protein purification...

(6) I would suggest to the authors that at least a double NXPE1/CASD1 KO is investigated to get more of an idea of how this new enzyme fits in to the overall pathway.

Reviewer #3

(Remarks to the Author)

Sialic acids attached to proteins and lipids can be modified by 9-O-acetylation. This post-glycosylation modification significantly alters properties of the Sia, including the abrogation or acquisition of the ability of specific lectins to bind, and alterations in physiochemical properties. After many attempts to identify an enzyme in mammals, finally, in 2015 one enzyme, CASD1 was reported that catalyzes 9-O-acetylation.

In normal colonic tissue mPAS is known to stain sialic acids present on goblet cells. Apparently, it does not stain 9-O-acetylated Sias. Making use of the finding that mPAS staining follows Hardy-Weinberg patterns of inheritance the authors used linkage analysis to locate the gene responsible for loss of mPAS reactivity. A marker, rs661946, correlated perfectly with lack or presence of mPAS staining, and is located in the promoter region of NXPE1 [also known previously as FAM55A], a gene of unknown function. The SNP associated with rs661946 is a C/T, T/T or C/C. The investigators overexpressed NXPE1 in Jurkat cells, correlating its expression with reduced mPAS signals and SIGLEC-15 binding. CRISPR-mediated exchange of a T to a C in rs661946 in a colon cancer cell line originally homozygous for a T decreased mPAS staining. Finally, an in vitro assay using the extracellular domain and bacterially expressed protein, they showed that NXPE1 indeed has the ability to transfer acetate to CMP-Neu5Ac in the presence of acetyl-CoA.

Since 2015, other SOATs had not been identified. Approaches based on sequence homology to CASD1 did not yield candidates. Therefore, this is a remarkable study that took a novel approach to identify a second enzyme that acts as a SOAT in human cells.

Individual experiments presented here are indirect, whereas other approaches could have more directly supported the presence of 9-O-acetylated Sia. However, the total body of evidence as outlined in the discussion warrants the main conclusion of this study, that they have indeed identified a second human SOAT. In view of the biological and biochemical importance of 9-O-acetylation of Sia on glycoproteins and glycolipids, the expression of NXPE1 in colon is likely to set the stage for further investigations of the enzyme in mucosal gut immunity, colon cancer and inflammatory bowel conditions, as noted by the investigators.

Major specific comments:

1. As written, the paper both makes use of the fact that, as well as presumably shows for the first time [because no literature references are provided] that mPAS staining of and SIGLEC-15 binding to Sia is inhibited by Sia 9-O-acetylation. The presence of 9-O-Sia is however not directly shown either by biochemical [glycoproteomics] analysis or by use of lectins such as that of the crab Cancer antennarius that specifically detect such structures.

2. Detection of protein expression of NXPE1 in this study, and results in Fig. 3A, B and Fig. 4E for NXPE1 depend on the exclusive use of IHC and a single antibody from a commercial source. This antibody does not appear to have been used in other publications and there is no indication of quality control for example using a knockout cell line. This antibody is listed as usable for western blotting. Have the investigators shown that the antibody detects one single protein band in for example the Jurkat cells with OE of NXPE1, and with no signal in parental Jurkat cells?

3. Fig. 4B. The authors expressed NXPE1 in pLenti-C-mGFP-P2A-Puro and in the Methods section, they describe "Selection for 14 days was performed with 1 µg/mL of puromycin; Cells were washed and then monitored for GFP and target gene expression". However, in the main text and figure the Jurkat cells are described as a 'pool of cells'. Thus it is not clear what percentage of cells were successfully transduced. This is an issue for interpretation of the ICC. The resolution of the images in Fig. 4B is very poor and does not allow visualization of individual Jurkat cells. Nonetheless it appears that sTn shows a varying degree of intensity of staining, in parental Jurkat with some cells showing high levels and others not. Similarly the NXPE1 OE cells show overall reduced staining but it is not possible to distinguish individual cells. Since the sTn is a surface structure why didn't the investigators use flow to quantitate and compare signals on these cells?

4. Fig. 4E. LS180 is a colon adenocarcinoma cell line. Presumably the three different panels with T/T, C/T and C/C show homogenous populations of cells. What is the explanation that T/T cells, with relatively low NXPE1 expression, only show

few cells positive for mPAS staining?

Minor comments:

Line 28 "mPAS staining is known to reflect sialic acid O-acetylation" is ambiguous. The authors should cite the papers that have shown this, and/or clearly state: mPAS stains sialic acids and staining has been shown not to occur when the Sia are modified by O-acetylation.

Line 134. IHC showed cytoplasmic granular immunostaining. However the protein is repeatedly referred to as having an extracellular domain, which also contains the enzymatic activity. Extracellular indicates outside the cell. Do the authors think this enzyme is active on the cell surface?

Lines 62 and Line 140: "These observations support the idea that an unknown autosomal dominant human gene is responsible for the differential O-acetylation that underlies mPAS staining" "Heterozygotes show NXPE1 staining but no mPAS". This result would be consistent with the 'dominant' effect of NXPE1 on mPAS staining, that is, production of protein from one allele is sufficient to 9-O-acetylate all available Sia substrates in these cells. This could be mentioned in the discussion.

Line 144. "patients" but these are presumably normal samples?

Line 151: "O-acetylation modifications are capable of controlling sialic acid-SIGLEC interactions". This has been shown for some Siglecs. Has this ever been shown directly for Siglec-15? This should be clearly stated.

Line 169.. "and has very low expression of NXPE1" Does ref 55 provide information on the expression of NXPE1 in Jurkat cells? If not, what is the lack of expression statement based on?

Line 174 an anti-sialyl-Tn antibody 'sensitive to acetylation state'. What does that mean? Presumably the antibody does not bind when the sialyl-Tn is acetylated?

Lines 184-189 It is not clear what the significance is of Fig. 4C is. If the allele with a T has loss of transcription of NXPE1 in goblet cells how can there be "more expression from the T, rather than the C allele" in testis?

Line 251: "preference for SIGLEC-15 to bind NXPE1 modified proteins". This statement is the opposite of the message in this paper. "NXPE1 modified proteins" are presumed to be sialoglycoproteins with 9-O-acetylation?

Line 262-266 The authors wrote: "Genome-wide association studies have shown that the genomic region containing NXPE1 is linked to inflammatory bowel disease. It is possible that NXPE1, or the closely positioned NXPE4, are responsible for this linkage. If so, then one could imagine that NXPE enzymatic inhibitors could be used to treat a subset of patients with this disease." The information that there is a genetic linkage of NXPE1 to inflammatory bowel disease does not include information if there is overexpression or loss of expression of NXPE1. If there is loss of expression inhibitors would not be useful.

Line 263. This gene was also known as FAM55A. Kume et al [PMID 24687888] in fact identified this protein using membrane protein proteomics as expressed in colorectal cancer with/without metastasis, although the difference was not statistically significant.

Line 267-267 "Second, loss of 11q23 is found in more than 20% of colorectal cancers. Approximately 50% of these patients will be C/T heterozygotes at 268 rs661946". This statement does not make sense in the absence of information about general allele frequency of CT heterozygosity.

Figure 5. What are the purple cells and the lines at their tops?

Figure S2. The protomer and direction of transcription of NXPE1 need to be included.

Version 1:

Reviewer comments:

Reviewer #1

(Remarks to the Author)

This resubmission is substantially strengthened by the addition of new data that further delineates the biochemical activities of NXPE1 and discloses the unanticipated relationship between NXPE1 SNPs and the complex splicing pattern of NXPE1. The manuscript is also strengthened by targeted mutagenesis studies that support the previously structurally inferred location of the NXPE1 SOAT catalytic site.

The manuscript is also strengthened by clarification as to the cellular location of NXPE1 and by changes in wording that remove previous sources of potential confusion by the reader.

Taken in toto, the manuscript provides a convincing body of data that NXPE1 is a new human SOAT, a novel finding of significance to biochemistry and to colon physiology.

This reviewer has no further reservations regarding the study.

Reviewer #2

(Remarks to the Author)

The authors have undertaken a fairly comprehensive revision and while not everything has been addressed, I believe that the manuscript is now in an acceptable form adding substantial new and exciting knowledge in the field of sialoglycobiology.

Reviewer #3

(Remarks to the Author)

The reviewers have addressed all of my remarks.

REVIEWER COMMENTS

Reviewer #1 (Remarks to the Author):

Lee et al.

NXPE1 alters the sialoglycome by acetylating sialic acids in the human colon.

This is an elegant, complete, and convincing study that solves a long standing open biological question of the basis of acetylation of sialic residues on mucins and other glycoproteins, particularly in the gut. The complementary utilization of population genetics, molecular biology, structural biology, and biochemical techniques convincingly establishes that NXPE1 is responsible for acetylation of sialic acid residues, and strongly suggests similar activity will be demonstrated by other members of the gene family. As pointed out in the discussion, there are several significant translational implications of the finding for identifying new therapeutic targets for IBD and colorectal cancer.

We thank the reviewer for their positive evaluation of our work.

Some minor comments are:

1 . It would perhaps merit a comment as to what the LD between rs661646 and the SNPs assessed in the QTL analysis, and the effect size seen in the QTL model (-0.2 per allele in a linear model of the transverse colon), suggest as to the effect size of the rs661646 alleles.

We thank the reviewer for this suggestion which has prompted us to more thoroughly explore the variation in transcripts levels and splicing, both of which show tissue specific patterns and QTL effects. We have now included expanded material on this in Figure S8 and 4C. This analysis revealed complex slicing with multiple isoforms. Most importantly, we independently confirmed the allelic effects on expression by performing targeted RNA-sequencing on NXPE1 colorectal fresh frozen human samples, and looked at expression of two coding SNPs in NXPE1. Depending on the transcript regions scored, there appears to be a ~25% (rs10891692) to 50% (rs524911) decrease in the number of RNA transcripts coming from the ALT (T) allele. Given the multiple isoforms, complex splicing, and multiple promoters, the ultimate effects were best described by our knock-in experiments (Figures 4D-F).

2. It would be of interest to show loss of mPAS staining on NXPE1 transfected Jurkat cells, or to comment on why this was not assessed. Given the completeness of the study, the reviewer imagines there was some technical issue with the mPAS assay on these cells.

We previously noted that mPAS staining occurs within highly-structured colon crypts. Given (i) the lack of similar structural features on leukemic Jurkat cells and (ii) likely differences in glycan signatures between leukemic and colonic cells, we expected mPAS staining to work sub-optimally on Jurkat cells. Nonetheless, we agreed with the reviewer that this could be informative and attempted to perform mPAS staining on NXPE1 transfected Jurkat cells. For this staining experiment, we used the same fixed Jurkat cell line blocks from Figure 4B in the manuscript and followed the mPAS staining protocol detailed in our manuscript. Unfortunately, staining was barely observable in both wildtype Jurkat and NXPE1-overexpressed Jurkat cells, likely due to technical limitations of mPAS as stated above and alluded to by the reviewer.

3 . It may be of interest to show an alignment or spatial model that displays the glycine and asparagine residues that are noted as lacking in NXPE1, but as being characteristic of SOATs from other species.

We've included some characteristic SOAT's from two bacterial species in the known SGNH/GDSL hydrolase family in figure 2.

4. It may prevent misunderstanding to reword:

"The preference for SIGLEC-15 to bind NXPE1 modified proteins suggests that these protein targets contain Ser/Thr-GalNac linked sialic acid at the α 2-3 or α 2-6 position." to clarify that SIGLEC-15 binds to the same proteins that can be modified by NXPE1 (as SIGLEC-15 binding is actually blocked from binding to proteins that have been modified/acetylated by NXPE1).

We thank the reviewer for pointing out this error in wording. We have modified the statement in lines 282-283 of the Discussion to the following:

"The preference for SIGLEC-15 to bind NXPE1 unmodified proteins suggests that these protein targets contain Ser/Thr-GalNac linked sialic acid at the α 2-3 or α 2-6 position."

Reviewer #2 (Remarks to the Author):

The authors present an interesting multidisciplinary study that suggests the identify a putative sialic acid O-acetyltransferase (SOAT) NXPE1 in the colon. The authors utilise a well-established mild periodic acid Schiff staining (mPAS) that can distinguish, in part, between O-acetylated (mPAS negative) and non-O-acetylated (mPAS positive) sialic acids in tissues.

To the best of my knowledge, the only characterised human sialic acid O-acetyltransferase (SOAT) is CASD1, and consequently the present study may bring to light new information in a very important area of sialoglycobiology and cancer.

We thank the reviewer for their overall positive evaluation of our work and for recognition of its potential significance in human disease.

Some General comments: (1) PAS is not selective - so the cells could be filled up with glycogen and they would still turn red. Was the staining done with a preliminary distase treatment? Glycogen metabolism is known to be altered in colon cancer too. A glycomic profile of the cells allegedly expressing Neu5,9Ac2 would add significant strength to the study and in my view is more reliable than PAS.

We agree with the reviewer that PAS staining is not selective. During development of our mild PAS staining technique we were careful to optimize our staining times to avoid overstaining. We did not incorporate a distase treatment. We acknowledge that additional characterization of associated sialoglycans could be informative; however, we believe that such experiments are beyond the scope of our work as we do not have expertise in these techniques in our lab. We would be excited to see follow-up work that build on these findings and the tools provided within the manuscript.

(2) The authors refer to NXPE1 as 'extracellular' - implying CMP-Neu5Ac and AcetylCoA should be found in the extracellular environment in order to work - but then it is stated that NXPE1 is in the intra-endoplasmic reticulum/golgi space . Further histo for colocalization studies are required to better characterise the enzyme's location.

We thank the reviewer for pointing out this inconsistency in wording. Based on the structure of NXPE1, we predict it to be localized to the membrane, with a single pass (AA 1-59) hydrophobic transmembrane domain and a larger (AA 60-547) enzymatically active extracellular domain. We acknowledge we have not done staining for colocalization to confirm the intracellular location. To avoid confusion we've removed lines 125-128 from our initial manuscript submission. Furthermore, we've included a change on line 136 to reflect that the IHC staining pattern for NXPE1 is cell surface and not cytoplasmic as originally put in error.

(3) Chromatograms - ABSORBANCE was measured at 373 nm (?) There is an evident 'bump' following the Neu5Ac peak that appears to correspond to Neu5,7Ac2 and so perhaps NXPE1 produces this product or both it and Neu5,9Ac2. Further functional characterisation is required and I would recommend evaluation of deoxy-Neu5Ac analogues to assess what the enzyme really does and/or spend some time to try to elucidate the real mechanism ?

We agree with the reviewer that we have yet to fully uncover the function or mechanism of NXPE1's enzymatic activity. Baumann and colleagues previously demonstrated that the only other known human

sialic acid O-acetyltransferase CASD1 is capable of transferring acetyl groups via a covalent acetyl-enzyme intermediate.

We attempted to find deoxy-Neu5Ac analogues as the reviewer suggested, but we were only able to find 2,3-dehydro-2-deoxy-Neu5Ac, also known as DANA. This analogue only has a loss of its hydroxyl group at position 2, where acetylation is typically not implicated, and this site is not expected to be the target of NXPE1 based on our biochemical studies. Generation of more relevant deoxy-analogues is unfortunately out of our expertise and we were not able to locate it commercially.

To address this referee's comments we performed additional experiments to explore mechanistic features of NXPE1 by demonstrating that acetyl-CoA is necessary in NXPE1-mediated sialic acid modification. Removal of acetyl-CoA from the reaction mixture led to a complete absence of modified sialic acid, suggesting that the acetyl group from acetyl-CoA plays a necessary role in NXPE1-mediated sialic acid modification. Furthermore, we now experimentally established that mutating the predicted catalytically active serine in NXPE1 to an alanine (S355A) abrogates NXPE1's ability to acetylate sialic acids. Similar mutation of a spatially proximal aspartic acid residue to an alanine (D526A) led to a reduction in NXPE1-mediated O-acetylation but not a complete abrogation as with S355A. Altogether, we believe that this new set of data provides important mechanistic insights and suggests mechanistic similarities between NXPE1 and CASD1. We've included these new results in **Figure 5 and S10**, both in the manuscript and shown below:

(4) The observation that this enzyme doesn't accept Neu5Gc as a sialic acid could be true - but it's puzzling considering colon cancer does express Neu5Gc - aka did they feed Neu5Gc to the enzyme? How did the authors prepare CMP-Neu5Gc..? It is stated that "N-glycolylneuraminic acid (Neu5Gc) is a sialic acid not found in humans and it's used as a control". Which is at least partially wrong, as we know that it is found BUT not synthesised.

We thank the reviewer for these suggestions.

1. We would like to note we never attempted to treat Neu5Gc with our recombinant NXPE1 protein, but instead used purified Neu5Gc molecule (Ludger/QA-Bio, Cat #CM-NEU-GC-01) as a control for our sialic acid reference panel (Agilent, Cat #GKK-407). Therefore, it is not known whether the enzyme accepts Neu5Gc as a substrate or not. To address this concern, we obtained commercially-available 82.2% pure CMP-modified Neu5Gc from Chemily Glycoscience (Cat #SN02020) and repeated our biochemistry assay using CMP-Neu5Gc as the sialic acid substrate. We observed a noticeable and significant peak at ~5.9 minutes (where unmodified Neu5Gc is expected to run). We also observed a minor peak at ~8.8 minutes, which could be the result of an acetylation modification to Neu5Gc. Nonetheless, given the limited magnitude of the modified peak, we believe that NXPE1 does not readily accept CMP-Neu5Gc as a preferred substrate and its activity instead largely pertains to CMP-Neu5Ac. We have now included the below figures as part of **Figure S10** in the manuscript.

2. We thank the reviewer for correcting our error in wording, we have modified the above statement, present in the figure description to Figure S10, to the following:

“N-glycolylneuraminic acid (Neu5Gc) is a sialic acid that is not synthesized in humans and is shown as a control.”

(5) There is no information regarding protein purification...

We thank the reviewer for pointing this out. We have now included a thorough description in our Methods section (lines 514-528) to include the following details about protein purification:

“Expression construct and protein expression was performed by GeneArt (ThermoFisher Scientific). Briefly, expression constructs for the predicted extracellular domain of NXPE1 (AA 60-547) or mutant variants of NXPE1 (NXPE1^{S355A} and NXPE1^{D526A}) were synthesized as human codon-optimized gene fragments and cloned into a pcDNA3.4-TOPO vector backbone with an IL2 secretion signal followed by a 6x histidine tag and GGGG linker, then NXPE1 protein sequence. NXPE1 expression plasmid DNA was purified from transformed DH10B E.coli competent cells through ThermoFisher Scientific’s GeneArt protein expression. The plasmid created above is stored and available at Addgene (Watertown, MA) under accession 226841. Recombinant NXPE1 was expressed by GeneArt (ThermoFisher) in Expi293 cells and purified using a HisTrap FF 5ml (Cytiva). The following buffers were used: wash buffer (20 mM imidazole, 500 mM NaCl, 25 mM Tris HCl pH 7.4) and elution buffer (500 mM imidazole, 500 mM NaCl, 25 mM Tris HCl pH 7.4). Recombinant NXPE1 was stored at -80°C in PBS until use.”

(6) I would suggest to the authors that at least a double NXPE1/CASD1 KO is investigated to get more of an idea of how this new enzyme fits in to the overall pathway.

We believe that NXPE1 and CASD1 function independently of one another, as suggested by the lack of concordance between mPAS, NXPE1 and CASD1 staining exemplified in **Figure 3B**. CASD1 has previously been shown to weakly modulate binding of SIGLEC-9 and SIGLEC-11 to cells (PMID 34192335). We purchased commercially available recombinant SIGLEC-9 (R&D Systems, Cat #1139-SL) and SIGLEC-11 (R&D Systems, Cat #3258-SL) to perform similar binding studies on rs661946-genotyped colon tissue. In our hands, NXPE1 status did not seem to correlate with either SIGLEC-9 or SIGLEC-11 binding, as exemplified below, suggesting that there are independent cellular outcomes to NXPE1 and CASD1’s sialic acid O-acetyltransferase activities.

Nonetheless, we agree with the reviewer that further evaluation of the potential relationship between NXPE1 and CASD1 could be valuable, and we are excited to see such work in future studies. The following statement has been added to lines 279-281 of the Discussion to acknowledge this:

“The specifics of how the acetyl-transfer is performed, the location on sialic acid targeted by NXPE1, and the potential relationship between CASD1 and NXPE1 remains to be investigated.”

Reviewer #3 (Remarks to the Author):

Sialic acids attached to proteins and lipids can be modified by 9-O-acetylation. This post-glycosylation modification significantly alters properties of the Sia, including the abrogation or acquisition of the ability of specific lectins to bind, and alterations in physiochemical properties. After many attempts to identify an enzyme in mammals, finally, in 2015 one enzyme, CASD1 was reported that catalyzes 9-O-acetylation.

In normal colonic tissue mPAS is known to stain sialic acids present on goblet cells. Apparently, it does not stain 9-O-acetylated Sias. Making use of the finding that mPAS staining follows Hardy-Weinberg patterns of inheritance the authors used linkage analysis to locate the gene responsible for loss of mPAS reactivity. A marker, rs661946, correlated perfectly with lack or presence of mPAS staining, and is located in the promoter region of NXPE1 [also known previously as FAM55A], a gene of unknown function. The SNP associated with rs661946 is a C/T, T/T or C/C. The investigators overexpressed NXPE1 in Jurkat cells, correlating its expression with reduced mPAS signals and SIGLEC-15 binding. CRISPR-mediated exchange of a T to a C in rs661946 in a colon cancer cell line originally homozygous for a T decreased mPAS staining. Finally, an in vitro assay using the extracellular domain and bacterially expressed protein, they showed that NXPE1 indeed has the ability to transfer acetate to CMP-Neu5Ac in the presence of acetyl-CoA.

Since 2015, other SOATs had not been identified. Approaches based on sequence homology to CASD1 did not yield candidates. Therefore, this is a remarkable study that took a novel approach to identify a second enzyme that acts as a SOAT in human cells.

Individual experiments presented here are indirect, whereas other approaches could have more directly supported the presence of 9-O-acetylated Sia. However, the total body of evidence as outlined in the discussion warrants the main conclusion of this study, that they have indeed identified a second human SOAT. In view of the biological and biochemical importance of 9-O-acetylation of Sia on glycoproteins and glycolipids, the expression of NXPE1 in colon is likely to set the stage for further investigations of the enzyme in mucosal gut immunity, colon cancer and inflammatory bowel conditions, as noted by the investigators.

We appreciate the reviewer's overall positive evaluation of our work.

Major specific comments:

1. As written, the paper both makes use of the fact that, as well as presumably shows for the first time [because no literature references are provided] that mPAS staining of and SIGLEC-15 binding to Sia is inhibited by Sia 9-O-acetylation. The presence of 9-O-Sia is however not directly shown either by biochemical [glycoproteomics] analysis or by use of lectins such as that of the crab Cancer antennarius that specifically detect such structures.

We thank the reviewer for this suggestion. Our team unfortunately does not have sufficient expertise nor technical equipment for performing glycoproteomic analysis and therefore anticipate significant challenges in performing this work. Instead, we attempted to obtain crab Cancer antennarius lectin as suggested by the reviewer from the following commercial sources: EY Laboratories (Cat #F-7201-1), Creative Biomart (Cat #Lectin-4001C), and CD BioGlyco (Cat #XLC-15-12Q and #XLC-15-04Q). Despite repeated attempts of purchase, all three sources were unable to provide us with this product.

To try and address these comments as best we could we performed similar binding assays on colon tissue using other lectins and glycan-binding domains, but none were able to specifically recognize the acetyl mark as expected from their predicted binding partners.

In lieu of this, we have removed any definitive claim within the manuscript text of NXPE1 mediating O-acetylation at the 9th carbon position, and we instead now frame our findings as a suggestion for the location of NXPE1-mediated O-acetylation. For instance, we modified line 274 of our Discussion to read:

“(ix) NXPE1 protein can acetylate sialic acids in vitro (Figure 5A).”

2. Detection of protein expression of NXPE1 in this study, and results in Fig. 3A, B and Fig. 4E for NXPE1 depend on the exclusive use of IHC and a single antibody from a commercial source. This antibody does not appear to have been used in other publications and there is no indication of quality control for example using a knockout cell line. This antibody is listed as usable for western blotting. Have the investigators shown that the antibody detects one single protein band in for example the Jurkat cells with OE of NXPE1, and with no signal in parental Jurkat cells?

We appreciate this referee’s commitment to ensuring strong specificity of our IHC staining. We have attempted to use this antibody without success by western blot. To address this concern further we repeated the staining with a second unrelated antibody, 1:100 dilution; catalog# HPA049133, Lot# R59143, Sigma-Aldrich, also used by the Human Protein Atlas for their work. The results of the staining between both antibodies is identical. We’ve included the information for this second antibody in the methods on line 472-473 and 478-480, and attached a representative image of the results below. We’ve also made mention of confirming these results with 2 unique antibodies in the discussion on line 256.

3. Fig. 4B. The authors expressed NXPE1 in pLenti-C-mGFP-P2A-Puro and in the Methods section, they describe “Selection for 14 days was performed with 1µg/mL of puromycin; Cells were washed and then monitored for GFP and target gene expression”. However, in the main text and figure the Jurkat cells are described as a ‘pool of cells’. Thus it is not clear what percentage of cells were successfully transduced. This is an issue for interpretation of the ICC. The resolution of the images in Fig. 4B is very poor and does not allow visualization of individual Jurkat cells. Nonetheless it appears that sTn shows a varying degree of intensity of staining, in parental Jurkat with some cells showing high levels and others not. Similarly the NXPE1 OE cells show overall reduced staining but it is not possible to distinguish individual cells. Since the sTn is a surface structure why didn’t the investigators use flow to quantitate and compare signals on these cells?

We thank the reviewer for this suggestion. We agree that such quantitation could be valuable and resurrected the same NXPE1-overexpressing Jurkat cell line as used in **Figure 4** of the manuscript. We stained these cells with anti-sTn antibody (Abcam, Cat #ab115957) at a concentration of 2µg/mL followed by an anti-mouse IgG-Alexa Fluor 647 secondary antibody (Cell Signaling, Cat #4410) at a concentration of 10µg/mL. Interestingly, but perhaps unsurprisingly based on Figure 4B, we noted a bimodal staining pattern as shown below. NXPE1-overexpressing Jurkat cells demonstrated marked reduction in anti-sTn staining, similar to that shown by **Figure 4B**. The figure shown below has now been added to our manuscript as **Figure S7**. Figures previously noted as Figure S7-9 have been appropriately re-numbered.

4. Fig. 4E. LS180 is a colon adenocarcinoma cell line. Presumably the three different panels with T/T, C/T and C/C show homogenous populations of cells. What is the explanation that T/T cells, with relatively low NXPE1 expression, only show few cells positive for mPAS staining?

We observed that mPAS staining was never perfectly recapitulated in cell line culture, likely owing to the lack of key extracellular structural features that enable proper mPAS staining in colorectal tissue samples (Figure S8 in the manuscript). This leads to the sporadic mPAS staining patterns *in vitro* that the reviewer noted.

Minor comments:

Line 28 “mPAS staining is known to reflect sialic acid O-acetylation” is ambiguous. The authors should cite the papers that have shown this, and/or clearly state: mPAS stains sialic acids and staining has been shown not to occur when the Sia are modified by O-acetylation.

We have previously referred readers to references 5-15 in line 53 of the manuscript text which illustrate the use of mPAS staining in this context. As the reviewer suggested, we have edited our manuscript text in lines 28-29 to make this clearer:

“mPAS stains sialic acids except when these glycans are modified by O-acetylation, but a full accounting of the genes contributing to sialoglycan acetylation is incomplete.”

Line 134. IHC showed cytoplasmic granular immunostaining. However the protein is repeatedly referred to as having an extracellular domain, which also contains the enzymatic activity. Extracellular indicates outside the cell. Do the authors think this enzyme is active on the cell surface?

This was indeed an error in our writing and after reviewing the NXPE1 staining more closely have change this to read “membrane granular immunostaining.” The staining is mostly around the cell surface, not in the cytoplasm.

Lines 62 and Line 140: “These observations support the idea that an unknown autosomal dominant human gene is responsible for the differential O-acetylation that underlies mPAS staining”
 “Heterozygotes show NXPE1 staining but no mPAS”. This result would be consistent with the ‘dominant’ effect of NXPE1 on mPAS staining, that is, production of protein from one allele is sufficient to 9-O-acetylate all available Sia substrates in these cells. This could be mentioned in the discussion.

We agree with the reviewer that this is an intriguing finding. We have modified lines 207-210 in our manuscript text to further emphasize this point:

“A robust increase in NXPE1 protein and decrease in mPAS staining was observed in the engineered lines with T/T to C/T or C/C change, an observation further supporting the autosomal dominant behavior of NXPE1 (Figures 4E and F).”

Line 144. “patients” but these are presumably normal samples?

All tissue samples used in our study were from normal tissue or normal adjacent tumor colorectal tissue. This wording has now been modified to “Patient tissue samples...”

Line 151: “O-acetylation modifications are capable of controlling sialic acid-SIGLEC interactions”. This has been shown for some Siglecs. Has this ever been shown directly for Siglec-15? This should be clearly stated .

We thank the reviewer for pointing out this important observation. As the reviewer suggests, although other SIGLECs have previously been implicated to have modulated binding depending on O-acetylation marks (PMID 34192335), to the best of our knowledge there is no existing literature that suggests that the same occurs for SIGLEC-15. We have modified lines 155-158 in our manuscript text to reflect this distinction:

“Of twenty SIGLEC/sialic acid binding proteins tested, sialic acid-binding Ig-like lectin 15 (SIGLEC-15) consistently stained colon tissues in the same way as mPAS, suggesting a previously unknown role of O-acetylation modifications for SIGLEC-15 binding (Figures 3C and 3D, Table S4).”

Line 169..” and has very low expression of NXPE1” Does ref 55 provide information on the expression of NXPE1 in Jurkat cells? If not, what is the lack of expression statement based on?

Based on Human Protein Atlas data, NXPE1 is almost exclusively expressed in the gastrointestinal tract which would suggest its lack of expression in Jurkat cells, which were derived from a patient with T-cell leukemia. We demonstrate this with an anti-NXPE1 immunohistochemistry stain (**Figure S6** in our manuscript). We’ve indicated to readers to review **Figure S6** at line 174 in the manuscript.

Line 174 an anti-sialyl-Tn antibody ‘sensitive to acetylation state’ . What does that mean? Presumably the antibody does not bind when the sialyl -Tn is acetylated?

We thank the reviewer for pointing this out. We have modified our text as in lines 178-180 to lessen any potential confusion:

“... we then performed IHC using an anti-sialyl-Tn antibody and observed marked reduction in staining for Jurkat cells with NXPE1 overexpression, presumably due to NXPE1-mediated sialyl-Tn O-acetylation.”

Lines 184-189 It is not clear what the significance is of Fig. 4C is. If the allele with a T has loss of transcription of NXPE1 in goblet cells how can there be “more expression from the T, rather than the C allele” in testis?

We appreciate the reviewer’s comment which has prompted us to look more closely at the tissue and QTL related expression (Figure S8 and new figure 4C). We’ve elaborated as to what effect the genotype present in the SNPs noted in this study appear to have on expression and splicing based on data

available in the GTEx database, and now show them in these figures (4C and S8). The absolute level of expression in testis is less than that in transverse colon and most germane to the reviewer's comment, the isoforms expressed in testis appear to originate from a different promotor (Figure S8).

Line 251: "preference for SIGLEC-15 to bind NXPE1 modified proteins". This statement is the opposite of the message in this paper. "NXPE1 modified proteins" are presumed to be sialoglycoproteins with 9-O-acetylation?

We thank the reviewer for pointing out this error. We have modified the statement to read "...unmodified proteins"

Line 262-266 The authors wrote: "Genome-wide association studies have shown that the genomic region containing NXPE1 is linked to inflammatory bowel disease. It is possible that NXPE1, or the closely positioned NXPE4, are responsible for this linkage. If so, then one could imagine that NXPE enzymatic inhibitors could be used to treat a subset of patients with this disease." The information that there is a genetic linkage of NXPE1 to inflammatory bowel disease does not include information if there is overexpression or loss of expression of NXPE1. If there is loss of expression inhibitors would not be useful.

We agree with the reviewer that current literature is limited as to whether these diseases are associated with an up- or down-regulation of NXPE1 expression. We have modified our manuscript text as in lines 311-313 to reflect this:

"If so, then one could imagine that NXPE1-targeted therapies could be used to treat a subset of patients with this disease if NXPE1 is confirmed as a contributing factor."

Line 263 . This gene was also known as FAM55A. Kume et al [PMID 24687888] in fact identified this protein using membrane protein proteomics as expressed in colorectal cancer with/without metastasis, although the difference was not statistically significant.

We thank the reviewer for pointing out this study. As the referee suggests, we believe that there may be diagnostic and/or prognostic value to NXPE1 in patients with colorectal cancers. We demonstrated that NXPE1 expression determines binding of SIGLEC-15, a known immune suppressive receptor (PMID 30833750), to colon sialoglycans. Moreover, previous analysis has suggested a sequential reduction of NXPE1 expression along the colon mucosa-adenoma-carcinoma sequence (PMID 29659199), in addition to the association noted by the reviewer. We've included this recommended reference in line 310 of the Discussion.

Line 267 -267 "Second, loss of 11q23 is found in more than 20% of colorectal cancers. Approximately 50% of these patients will be C/T heterozygotes at 268 rs661946". This statement does not make sense in the absence of information about general allele frequency of CT heterozygosity.

We've edited this statement to read:

"Second, loss of 11q23 is found in more than 20% of colorectal cancers. Across the general population, with an allele frequency of 0.38, approximately 50% of these patients will be C/T heterozygotes at rs661946."

Figure 5 . What are the purple cells and the lines at their tops?

The purple cells are enteroendocrine cells and the lines represent neuropods. We've added a note in the figure legend describing these cells.

Figure S2. The promoter and direction of transcription of NXPE1 need to be included.

We thank the reviewer for this suggestion. Figure S2 has now been edited to include the promoter and direction of transcription for all genes.